# Improving Adversarial Training for Two-player Competitive Games via Episodic Reward Engineering

**Siyuan Chen**                                              *alsachai@g.ecc.u-tokyo.ac.jp*
*The University of Tokyo*

**Fuyuan Zhang** *                                           *fuyuanzhang@zju.edu.cn*
*Zhejiang University*

**Zhuo Li**                                                  *zhuoli.academic@gmail.com*
*Kyushu University*

**Xiongfei Wu**                                              *xiongfei.wu@uni.lu*
*University of Luxembourg*

**Jianlang Chen**                                           *chen.jianlang.396@s.kyushu-u.ac.jp*
*Kyushu University*

**Pengzhan Zhao**                                           *zhaopengzhan@hebtu.edu.cn*
*Hebei Normal University*

**Lei Ma**                                                   *ma.lei@acm.org*
*The University of Tokyo, University of Alberta*

**Jianjun Zhao**                                            *zhao@ait.kyushu-u.ac.jp*
*Kyushu University*

**Reviewed on OpenReview:** *https://openreview.net/forum?id=z4XtJWJC9K*

## Abstract

In recent years, training adversarial agents has become an effective and practical approach for attacking neural network policies. However, we observe that existing methods can be further enhanced by distinguishing between states leading to win or lose and encouraging the policy training by reward engineering to prioritize winning states. In this paper, we introduce a novel adversarial training method with reward engineering for two-player competitive games. Our method extracts the historical evaluations for states from historical experiences with an episodic memory, and then incorporating these evaluations into the rewards with our proposed reward revision method to improve the adversarial policy optimization. We evaluate our approach using two-player competitive games in MuJoCo simulation environments, demonstrating that our method establishes the most promising attack performance and defense difficulty against the victims among the existing adversarial policy training techniques. The source code is available at https://github.com/alsachai/episodic_reward_engineering.

## 1 Introduction

Prior work shows that deep reinforcement learning (DRL) policies can be vulnerable to adversarial attacks (Huang et al., 2017; Kos & Song, 2017). Most existing attacks on DRL policies are executed by

---

*Fuyuan Zhang is the corresponding author and is with the State Key Laboratory of Blockchain and Data Security, Zhejiang University, Hangzhou, China.

searching the adversarial examples and manipulating the environment (Huang et al., 2017; Kos & Song, 2017; Nguyen & Reddi, 2019). However, such adversarial examples may not be applicable in the real world (Gleave et al., 2020). Recently, training adversarial agents as attackers to DRL policies in two-player games has been proven effective and practical (Gleave et al., 2020; Wu et al., 2021; Guo et al., 2021; Bui et al., 2022). These kinds of attacks first reduce the two-player environments to single-player environments by fixing the victim agents, and then train the other agent to be an adversarial agent which can be trained by conventional single-agent policy training methods. Known as adversarial policy training, these attacks generate natural observations that are adversarial to the victim agents, achieving significant results.

While the aforementioned adversarial policy training methods have proven effective, there is still room to improve adversarial training by providing the training process with higher quality environmental rewards to learn. Our insight is simple: utilizing the adversarial agent's historical experiences to distinguish between states that lead to wins or loses, thereby assigning higher rewards to the winning ones and lower rewards to the losing ones. Based on this insight, we introduce a novel adversarial policy training approach that leverages the analysis of information from past episodes to assess game states and improve the environmental rewards with those assessments to assist the adversarial agent in achieving better performance.

In this paper, we propose an adversarial policy training method with reward engineering mechanism for two-player competitive games. Inspired by previous works on improving the rewards for policy learning using episodic control (Blundell et al., 2016; Pritzel et al., 2017; Li et al., 2023), our method develops a neural network-based episodic memory to store historical experiences and generate state evaluations corresponding to the game outcomes. We then design a conditional reward revision method to improve the environmental rewards based on those evaluations. In our experiments, we evaluate our method on two-player competitive games in MuJoCo domains (Todorov et al., 2012) and compare it with state-of-the-art adversarial policy training approaches (Gleave et al., 2020; Guo et al., 2021; Wu et al., 2021). Our experimental results show that our method establishes the most promising attack performance and defense difficulty.

In summary, this paper makes three contributions. First, we propose a novel adversarial policy learning framework with reward enhancement for two-player competitive games. Second, we introduce a neural network-based episodic memory that leverages historical experiences to evaluate states, along with a reward revision approach that incorporates these historical evaluations into environmental rewards. Third, our work demonstrates that by identifying and highlighting the winning states based on historical experiences with reward revision, adversarial agents can achieve higher win rates against fixed victim agents and may have the potential to possess the capability to explore a more effective strategy by integrating actions from past successful strategies to defeat the victim, which is demonstrated in one Mujoco game environment in our experiments.

## 2 Related Work

### 2.1 Adversarial Attacks on DRL Policies

Previous attacks against DRL policies mainly focus on manipulating the environment to fail the victim agents. One type of attack focuses on perturbing the victim's observations, forcing its policy network to output sub-optimal actions, and thus failed the victim agent (Russo & Proutiere, 2019; Sun et al., 2020; Zhang et al., 2021; Madry et al., 2018; Pattanaik et al., 2018; Pan et al., 2022; Zhao et al., 2020). Another kind of attack directly perturbs the trajectory of the victim, specifically the actions the victim agent takes (Lee et al., 2020; Pan et al., 2022) or the rewards it receives (Ma et al., 2019; Yang et al., 2019; Lykouris et al., 2021) to effectively attack the victim. However, the above attacks are argued to be unrealistic since the real-world environment cannot be manipulated (Gleave et al., 2020; Guo et al., 2021; Wu et al., 2021).

Unlike the above attacks, to simulate the real-world scenarios, Gleave *et al.* successfully trained adversarial agents by PPO algorithm (Schulman et al., 2017) in two-player competitive games under a strict zero-sum assumption and demonstrated the effectiveness of training adversarial agents against fixed black-box victims (Gleave et al., 2020). Wu *et al.* further improved the attack performance by exploring the minimal observation differences of the shared environment to maximize deviations of the victim actions (Wu et al., 2021). Guo *et al.* relaxed the zero-sum assumptions of previous works and demonstrated that such attack

could be achieved by maximizing the gap between the adversary and victim rewards which are approximated by observations of the adversarial agent (Guo et al., 2021). On the other hand, Bui *et al.* adopted imitators of the victim policies learned by imitation learning algorithms (*e.g.*, GAIL (Ho & Ermon, 2016)) to roll out the victim actions for the attacker and reached better performances (Bui et al., 2022). However, this attack is based on the specification that the victim's actions are visible and accessible to the imitators.

This paper adopts the same setting as (Gleave et al., 2020; Guo et al., 2021; Wu et al., 2021), wherein we have control solely over the adversarial agent and treat the victim agent as a black box, rendering its observations, actions, and rewards inaccessible. Meanwhile, unlike (Gleave et al., 2020; Guo et al., 2021; Wu et al., 2021), we concentrate on leveraging the historical experiences from adversarial agents to emphasize the winning states to improve the adversarial policy training. Our experiments in Section 4.2 show the effectiveness of such improvement.

## 2.2 Reward Engineering

Reward design plays a pivotal role in reinforcement learning, where poorly structured rewards can stall policy improvement. Consequently, various methods (Andrychowicz et al., 2017; Schaul et al., 2015) have been proposed to improve the reward signal. For example, Hindsight experience replay (Andrychowicz et al., 2017) sets an alternative goal for the state transitions of an episode to create a learning signal from failed attempts, and then modifies the rewards based on whether the alternative goal is reached.

In recent research, episodic control (Lengyel & Dayan, 2007) shows its effectiveness on modifying environmental rewards to address sample inefficiency in various tasks, such as multi-agent tasks (Zheng et al., 2021), model-based reinforcement learning (Le et al., 2021), and continuous control (Zhang et al., 2019; Kuznetsov & Filchenkov, 2021). Previous works primarily adopt a tabular episodic memory to save experiences of past scenes, leveraging the information gained during exploration and retrieving past experiences of similar scenes to expedite policy optimization (Blundell et al., 2016). This memory uses the state as a key and a measurement of that state (which can vary by method, e.g., a Q-value) as the value, storing these key-value pairs. Then, a distance-based analysis (*e.g.*, KNN) is adopted to retrieve a summary statistic of similar states from the episodic memory, and the retrieved statistic can be used to guide the reward engineering to improve the training process (Hansen et al., 2018).

This paper adopts episodic control as the base since it shares a similar empirical insight of using historical experience to guide learning. Unlike methods that directly add statistics from episodic memory to the reward, we apply a conditional revision aligned with outcome-aligned learning signals and rely on a neural episodic memory for efficiency. Compared with a state-of-the-art episodic control method NECSA (Li et al., 2023), our approach performs better with less time cost (See Section 4.3).

## 3 Methodology

In this work, we propose an adversarial training approach with reward engineering for training adversarial agents. Our method utilizes the conventional adversarial policy training framework and improves the rewards used for training. The workflow of our approach is shown in Figure 1. First, the adversarial agent interacts with the environment and gathers information (states, actions, rewards) from the episodes. This information is adjusted using our proposed reward revision method, and then saved in the experience storage (*e.g.*, replay buffer). We calculate the objective function based on the revised rewards sampled from the experience storage and update the agent's policy with the objective function. In the following, we mainly elaborate on our reward engineering method based on historical experiences analysis.

## 3.1 Adversary in Two-player Markov Game Environment

A two-player Markov game environment can be modeled as $E = (S, (A_\alpha, A_\nu), T, (R_\alpha, R_\nu))$. Here, we use $\alpha$ and $\nu$ to represent the adversary and victim respectively. $S$ represents a state set, and both $A_\alpha$ and $A_\nu$ are action sets. $T$ denotes a joint state transition function $T : S \times A_\alpha \times A_\nu \to \Delta(S)$, where $\Delta(S)$ is a probability

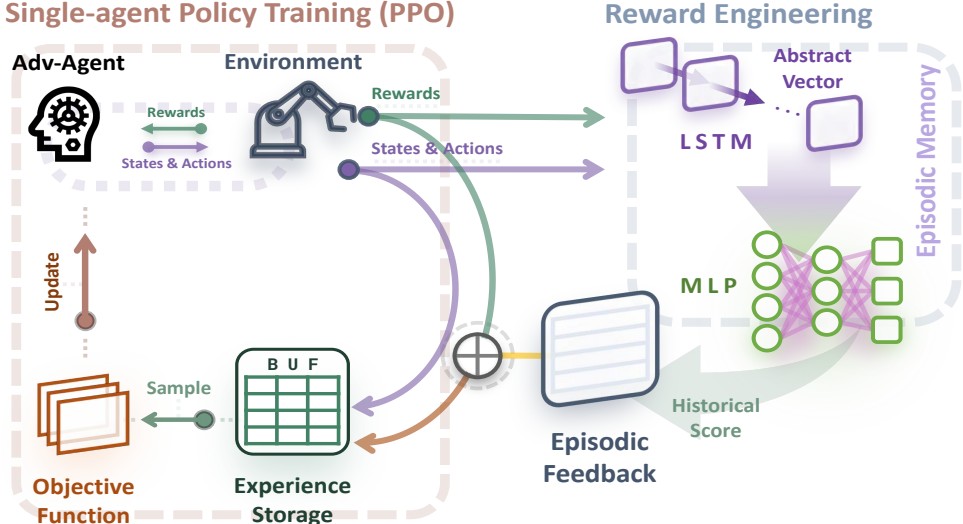

Figure 1: The workflow of our adversarial policy training. The process consists of four main steps: (1) The adversarial agent interacts with the environment and gathers episode information (states, actions, rewards). (2) The episodic memory collects the information to generate historical evaluation for states. (3) The original environment rewards are revised using this evaluation. (4) The agent's policy is updated based on the revised rewards.

distribution on $S$. The reward function $R_i : S \times A_\alpha \times A_\nu \times S \to \mathbb{R}$ depends on the current state, actions taken by both agents and the next state.

Gleave *et al.* discovered that by fixing the victim agents, the two-player competitive games can be reduced to single-player games and one agent can be trained as an adversarial agent with conventional single-agent policy training methods (*e.g.*, PPO) to attack the victim (Gleave et al., 2020). The training for the adversarial agent to defeat the fixed victim agent is called **adversarial policy training**. Under this setting, the two-player game can be regarded as a single-player MDP: $E_\alpha = (S, A_\alpha, T_\alpha, R'_\alpha)$ as the victim policy can be treated as a part of the environment. $S$ becomes the state set of the adversary and the state transition function and reward function change to $T_\alpha : S \times A_\alpha \times S \to \Delta(S)$ and $R'_\alpha : S \times A_\alpha \times S \to \mathbb{R}$.

**Our threat model** We use the same setting as Gleave *et al.* and assume that our threat model has control over the adversarial agent and black-box access to the information of the victim agent. The adversarial agent can only interact with the environment, which is the common practice in adversarial policy training. We also assume that the environment can only terminate with a terminal signal indicating three possible outcomes: adversary win, adversary loss, or tie.

## 3.2 Reward Revision Based on Historical Experiences

The environmental reward $R'_\alpha$ obtained by the adversarial agent only includes the evaluations from a single game and does not contain evaluations from the historical experiences. To enhance the adversarial policy training, our key insight is to integrate state evaluations from historical experiences into the rewards, thereby providing the adversarial policy with more comprehensive rewards for learning.

As a two-player Markov game is not deterministic, past episodes starting from the same state may include different state sequences. Therefore, we use state sequences, referred to as patterns, as the basis for performance evaluation in past episodes. Our choice of utilizing state sequence for performance evaluation aligns with existing research (Sutton & Barto, 2018; Li et al., 2023), which has proved that failures in adversarial games are often the result of a series of poor decisions rather than isolated states. Based on the qualification of these patterns from historical experiences, we can assign higher rewards to states that lead to good pat-

terns and lower rewards to states that lead to bad patterns, thereby integrating evaluations from historical experiences into the rewards.

### 3.2.1 Evaluation of Pattern Performance

As mentioned above, a pattern refers to a sequence of states within an episode. For a given episode $e$, a *k-step pattern* starting from time step $t$, denoted as $p_t$, refers to $k$ consecutive states in $e$, i.e., $p_t = s_t, \ldots, s_{t+k-1}$. For example, $p_1 = s_1, \ldots, s_k$ exemplifies a $k$-step pattern from the first state $s_1$. Notice that a state can be considered a special case of a pattern, specifically a 1-step pattern.

To evaluate the past performance of the patterns, we propose that a pattern can be considered **high-performing** if the past episodes containing this pattern result in more wins than losses. Conversely, a pattern can be considered **poor-performing** if the past episodes containing it result in more losses than wins. Based on this idea, we introduce a **historical score** for each pattern to quantify its past performance, defined as the average cumulative reward received in past episodes that include that pattern. The cumulative reward for an episode, which quantifies the adversarial agent's performance, tends to be higher in episodes where the agent wins than in those it loses. As the performance of adversarial agents improves, these cumulative rewards increase. Therefore, taking the average cumulative reward across all episodes that contain a particular pattern captures the pattern's historical effectiveness. During training, we store each pattern alongside the cumulative reward of its corresponding episodes in an episodic memory. The memory then returns the average cumulative reward as the historical score for that pattern. The historical score $h\_score(p_t)$ can be calculated as:

$$h\_score(p_t) = M(p_t) = \overline{\mathcal{R}_{p_t}}, \tag{1}$$

where $\overline{\mathcal{R}_{p_t}}$ is the average cumulative reward of past episodes containing $p_t$ and $M$ is an episodic memory we proposed to collect and analyze patterns in past episodes. In Section 3.3.1, we will further explain how we implement the episodic memory.

### 3.2.2 Conditional Reward Revision

Based on the historical scores of patterns, we incorporate the historical evaluations into rewards by reward revision. As revising rewards may lead to different optimal policies, inspired by potential-based shaping functions (Ng et al., 1999), we propose episodic feedback to keep the reward signal mean-zero to preserve the policy-optimality. It is defined as the difference between the historical score of a pattern and the average cumulative reward of all past episodes:

$$\delta(p_t) = h\_score(p_t) - \overline{\mathcal{R}}. \tag{2}$$

where $h\_score(p_t)$ is the historical score of pattern $p_t$ and $\overline{\mathcal{R}}$ is the average cumulative reward of all past episodes. The proof of the policy-optimality guarantee is provided in Appendix A.1. On the other hand, episodic feedback also contributes to the improvement of training. Semantically, $h\_score(p_t)$ represents the historical evaluation of one pattern while $\overline{\mathcal{R}}$ represents the agent's global average performance. Therefore, episodic feedback transforms the reward shaping signal from an absolute measure of quality into a relative one. Instead of just rewarding "good" patterns, it specifically rewards patterns that are better than the agent's current average performance. This provides a more focused and informative signal for credit assignment, guiding the policy to prioritize patterns that lead to more exceptional outcomes.

With the episodic feedback, we conditionally add the episodic feedback of a pattern to the reward of its initial state. Specifically, for an episode $e$, depending on the outcome of $e$ (whether the adversarial agent wins), we revise the rewards of the initial states of patterns in $e$ in two cases. Assume that $s_t$ is the initial state of pattern $p_t$ in episode $e$, whose reward is $r_t$, and $\delta(p_t)$ is the episodic feedback for $p_t$, the conditional

reward revision can be formulated as:

$$
\hat{r}_t = \begin{cases} r_t + \delta(p_t) \times \epsilon, & \text{if the adversarial agent wins} \\ & \quad and\ \delta(p_t) > 0, \\ r_t + \delta(p_t) \times \epsilon, & \text{if the adversarial agent loses} \\ & \quad and\ \delta(p_t) < 0, \\ r_t, & \text{otherwise,} \end{cases} \tag{3}
$$

where $\hat{r}_t$ is the revised reward and $\epsilon$ is a coefficient used to regulate the magnitude of encouragement and punishment. After the revision, we update $r_t$ with $\hat{r}_t$.

The reason we only revise the reward in the two cases mentioned above is that the revision must adhere to the win-loss rules of the two-player competitive game environment, even though this environment reduces to a single-player game environment during the training process. In a competitive game, we believe only states from a winning episode of the adversarial agent should be rewarded ($\delta(p_t) > 0$), while states from the losing episode should be penalized ($\delta(p_t) < 0$). We will further analyze other cases in Section 4.5.3.

### 3.3 Reward Revision Improving Adversarial Policy Training

#### 3.3.1 Neural Network-based Episodic Memory

As discussed in Section 3.2.1, we propose using an episodic memory to generate historical scores for patterns based on past experiences. Since reward revision with episodic memory adds to the total training time, we reduce its overhead by adopting a neural network-based architecture shown in Figure 1, where our episodic memory consists of an LSTM followed by a multi-layer perceptron (MLP). The LSTM encodes patterns into abstract vectors, and the MLP aggregates these encodings to produce historical scores. As reported in Section 4.2, the effectiveness shown in our experiments indicates that this simple architecture is sufficient for adversarial training in two-player competitive games.

During the training, new episodes are used to update the episodic memory. Assume a newly produced episode $e$ has a cumulative reward of $\mathcal{R}$ and contains the set of patterns $P$. The episodic memory $M$ is then trained to map all the patterns in $P$ to the same cumulative reward $\mathcal{R}$. Note that for a pattern in $P$, the learning target is the historical score of the pattern, which is not a fixed value and will change as new episodes occur. Then, during the reward revision, $M$ predicts historical scores for the patterns as Equation 1 to generate historical evaluations for reward revision. Details of our implementation of the episodic memory including the architecture, the forward process and the update process can be found in Appendix A.4.

#### 3.3.2 Group-based Episodic Feedback

In practice, we find that later-generated episodes exhibit higher win rates than earlier-generated episodes during the training process. To achieve better performance in finding the optimal policy, we compare the historical score of a pattern with the average cumulative reward from recently generated episodes when computing the episodic feedback. To achieve this, we divide the episodes into groups of size $n$, where $n$ is a hyper-parameter, and calculate the average reward of past episodes in the group. The average reward of the first $i$ episodes in group $m$ can be calculated by

$$
\overline{\mathcal{R}}_i^m = \frac{\sum_{j=1}^{j=i} \mathcal{R}_j^m}{i},\ 1 \le i \le n. \tag{4}
$$

Thus, we compute the episodic feedback of pattern $p_t$ in the $i$th episode of group $m$ by

$$
\delta(p_t) = h\_score(p_t) - \overline{\mathcal{R}}_i^m, \tag{5}
$$

which is implemented in our experiments.

#### 3.3.3 Adversarial Policy Training with Reward Revision

---

**Algorithm 1** Adversarial policy training with reward revision.

---

**Input**: $\mathcal{A}$: Adversarial Agent, $E$: Environment, $M$: Our episodic memory, $\mathcal{B}$: Experience Storage, $O$: Objective Function from the Fundamental Training Method

**Parameter**: $k$: Pattern Length, $n$: Group Size, $\epsilon$: Revision Coefficient

**Output**: $\mathcal{A}$: A Well-trained Adversarial Agent

---

1: **while** Training does not reach the maximum step **do**
2:     $\mathcal{A}$ interacts with $E$ and generate state $s$, action $a$, reward $r$.
3:     $S.add(s), A.add(a), R.add(r)$.
4:     **if** an episode ends **then**
5:         $P \leftarrow Pattern(k, S)$
6:         $R_{cum} \leftarrow Cumulative\_reward(R)$
7:         $M.update(P, R_{cum})$
8:         $H\_Score(P) \leftarrow M(P)$
9:         $m \leftarrow GroupIndex, i \leftarrow EpisodeIndexWithinGroup$
10:       $\overline{\mathcal{R}}_i^m \leftarrow Average\_Reward(i, m)$
11:       $\Delta(P) \leftarrow Episodic\_Feedback(H\_Score(P), \overline{\mathcal{R}}_i^m)$
12:       $R' \leftarrow Reward\_Revision(R, \Delta(P), \epsilon)$
13:       $\mathcal{B} \leftarrow S, A, R'$
14:       Clear $S, A, R$
15:     **end if**
16:     **if** Check\_Update() is true **then**
17:         $Experiences \leftarrow Sample(B)$
18:         $O(Experiences)$
19:         $\mathcal{A}.update(O)$
20:     **end if**
21: **end while**
22: **return** $\mathcal{A}$

---

We implement our policy training as Algorithm 1. First, we have the adversarial agent interact with the environment and generate states, actions and rewards (Line 2-3). When an episode ends, we extract patterns from the episode (In practice, we use sliding window) and calculate the cumulative reward of the episode (Line 5-6). We then update the episodic memory with the patterns and the cumulative reward (Line 7), and predict the historical score for each pattern with the episodic memory(Line 8). Subsequently, we calculate the average cumulative reward of the group with Equation 4 (line 9-11), and then utilize the average cumulative reward to calculate the episodic feedback for each pattern following Equation 5 (Line 12). With the episodic feedbacks, the rewards of the states could be revised by Equation 3 (Line 13) under the condition stated in Section 3.2.2. After the reward revision, we store the states, actions and the revised rewards into the experience storage (Line 14). When the update condition is satisfied (*e.g.*, storage is full), we will sample some data from the storage and calculate the objective function from our selected policy training method(e.g., PPO) (Line 17-19). The adversarial agent will be updated with the objective function (Line 20). Iteration will end when the maximum training step is reached.

## 4 Evaluation

### 4.1 Main Experiment Setup

In our main experiments, we evaluate our approach along two dimensions, attack effectiveness and defense robustness, by asking two questions: (Q1) Does it secure higher win and non-loss (win plus tie) rates? (Q2) How difficult is it for opponents to defend against the adversarial strategies it learns?

To answer Q1, we employ PPO (Schulman et al., 2017) as the fundamental single-agent policy training method to train adversarial agents against well-trained *Zoo* agents (Bansal et al., 2018). We take three state-of-the-art approaches (Gleave et al., 2020; Wu et al., 2021; Guo et al., 2021) as baselines. Notably, Gleave

*et al.* 's method incorporates PPO for adversarial policy training without modifying PPO's fundamental mechanisms, thus effectively serving as a PPO baseline. To ensure fair comparisons, we evaluate the methods using the same two-player competitive games in the MuJoCo robotics simulator: *YouShallNotPassHumans*, *KickAndDefend*, *SumoAnts* and *SumoHumans*, running 5 seeds per environment. In the following, we briefly introduce these four environments.

- *YouShallNotPassHumans* places two humanoid robots in a corridor. One agent, the runner, must traverse the hallway from the start line to a finish plane at the far end; the other, the blocker, aims to stop that traversal for the entire episode. All forms of physical contact are legally allowed. The game ends immediately when the runner crosses the finish lane (the runner wins) or when the fixed time limit is reached without a crossing (the blocker wins).

- *KickAndDefend* sets up a humanoid kicker, a humanoid defender, and a single soccer ball on a flat rectangular pitch. The kicker's sole objective is to propel the ball through a goal mouth, while the defender tries to block or clear it before that happens. Game concludes as soon as the ball fully crosses the goal line (the kicker wins) or when the time limit expires with no goal scored (the defender wins).

- *SumoAnts* drops two four-legged "Ant" robots onto a raised circular platform. Each agent tries to remain inside the ring while forcing the opponent off the edge. A match ends when (1) the timer expires with both ants still inside the circle (tie); (2) a robot leaves the ring without having been touched by its rival at that moment (a self-out, tie); (3) both robots exit in the same simulation step so the engine cannot determine who went out first (tie); and (4) one robot leaves the platform (the other one wins).

- *SumoHumans* follows the same ring-out rules as *SumoAnts* but swaps the quadrupeds for full 27-degree-of-freedom humanoids. The objective also remains the same as *SumoAnts*: stay on the platform and make the opponent step or fall outside.

For the hyper-parameters, we set the $\epsilon$ as 0.1, pattern length k as 3 and Group Size n as 100. Further hyper-parameter analysis is detailed in Appendix A.3.

To investigate Q2, we conduct two different defense robustness tests. (1) Victim retraining: we retrain the victim agents against the fixed adversaries and track whether the adversaries' win and non-loss rates decline as the victim adapts. (2) Masked observation defense: we employ masked victim agents proposed by Gleave et.al. (Gleave et al., 2020) as a simple defense method against adversarial training in two-player competitive games. In this experiment, the victim agent's observations of the adversarial agent are fixed, simulating limited perceptual capabilities to avoid adversarial actions. Together, these experiments allow us to evaluate the robustness and effectiveness of the adversarial strategies developed by our approach.

All experiments were conducted on a system running Ubuntu 20.04.4 with Intel Xeon E5-1650 v4 CPU, NVIDIA GeForce RTX 3090, and 128 GB of memory.

## 4.2 Main Results

The comparison of the win rates and non-loss rates between our approach and the baseline approaches are summarized in Figure 2. We can observe that our proposed method reaches 88% and 89% win rates and outperforms the baseline methods significantly in *YouShallNotPassHumans* and *KickAndDefend*. In *SumoHumans*, our method also surpasses all baselines. These results indicate that by leveraging historical experiences to highlight the high-performing states, our agents demonstrate higher sample efficiency and have more potential to discover effective adversarial policies to defeat victim agents. In *SumoAnts*, since the win rates of all agents against the victim are far below 50%, we use the non-loss rates to measure the effectiveness of our method. From Figure 2(b), we observe that in *SumoAnts*, the non-loss rate of our agent is still able to surpass that of the agent trained by Guo *et al.*, which exhibits the second highest non-loss rate.

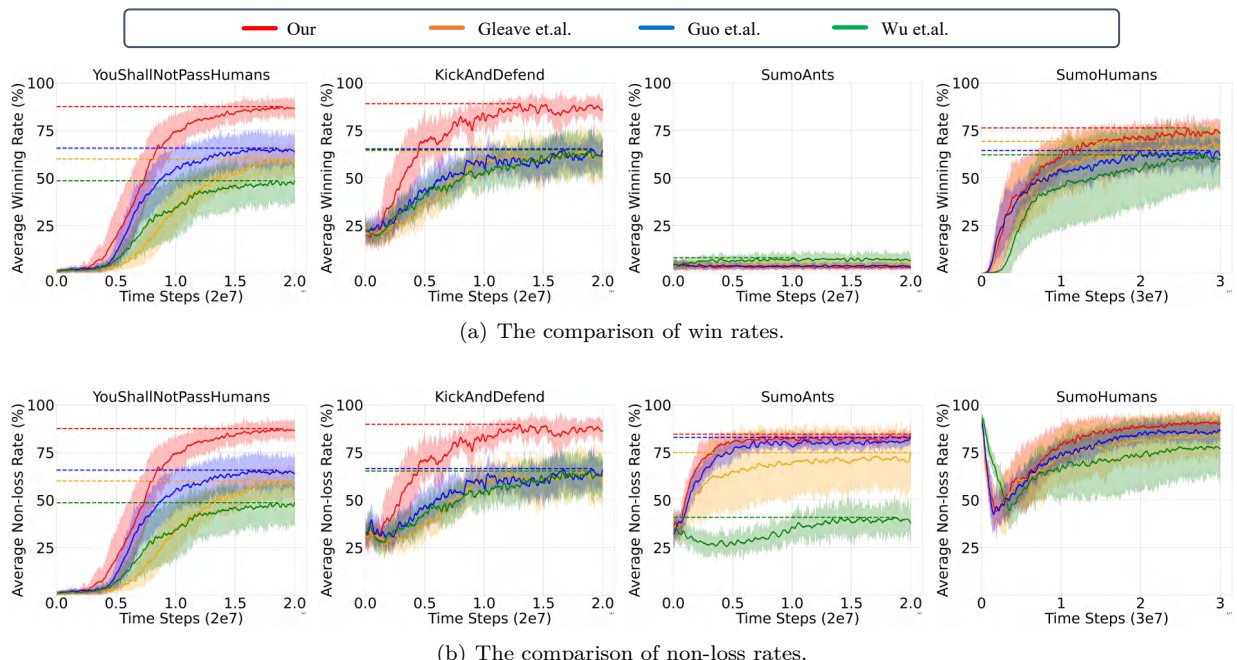

(a) The comparison of win rates.

(b) The comparison of non-loss rates.

Figure 2: The performance of our adversarial agents and baseline adversarial agents in each environment. Dashed lines represent the highest rates of each agent and non-loss rate is win rate plus tie rate. More details are shown in Table 5 and Table 6 in Appendix A.7.

We share the videos of agents trained by our approach and baseline approaches in Appendix A.6 and compare their behaviors. In *KickAndDefend* and *SumoHumans*, all the adversarial agents perform similar adversarial actions to trick the victim into performing abnormal behaviors. These results align with the conclusion in Gleave *et al.* that adversarial agents win by confusing the victim, instead of becoming a strong opponent. However, in *YouShallNotPassHumans*, while the three baseline agents attack the victim by convulsing on the ground, our agent simultaneously performs the adversarial action and obstructs the victim with a kicking. Kicking can be viewed as a non-adversarial action, illustrating that, by incorporating our improved rewards, the agent is not limited to a single winning strategy. Instead, it can draw on past successes, like a non-adversarial kicking action, to explore a **different but more effective** strategy. Additionally, in *SumoAnts*, agents trained by our approach and Guo *et al.* both jump out of the arena at the beginning since falling out of the arena without touching the opponent is considered a draw in this game, while agents trained by Gleave *et al.* and Wu *et al.* still fight with the victim. This suggests that same as Guo *et al.*, our agent is also capable of discovering and exploiting game imbalances.

To validate whether our approach is difficult to defend against, we conduct retraining experiments on the victim agents using the vanilla PPO algorithm. We report the results of our adversarial agent and baseline agents against the victim agents during the retraining in Figure 3. In *YouShallNotPassHumans*, our agent maintains a relatively high win rate during the retraining of victim agents, unlike baseline agents whose win rate quickly drops to a low level. In *KickAndDefend*, the win rates of our agent also decreases at a slower rate compared to the baseline agents. This indicates that our approach is more difficult to defend than baseline approaches.

In order to gain a better understanding of the effectiveness of our method, following the approach in Gleave *et al.*, we have our adversarial agents play games against masked victim agents, whose observation of the adversary's position is set to a static value corresponding to a typical initial position so that the adversarial actions may not be effective. We show the performances of our adversarial agents and baseline agents against masked victims in Table 1. We observe a significant decline in the non-loss rates of the three baseline agents, while our agent maintains a high non-loss rate against the masked victim in *YouShallNotPassHumans*. This could be attributed to the fact that our agent not only relies on adversarial actions to attack the victim,

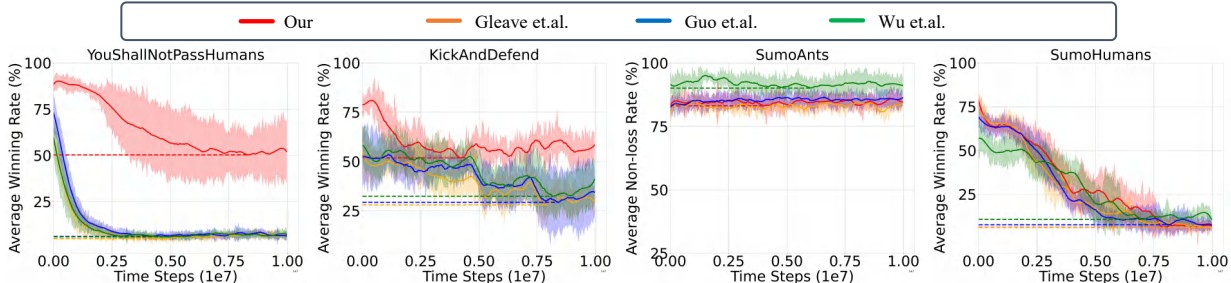

Figure 3: The comparison between our adversarial agent and baseline adversarial agents in each environment (Gleave et al., 2020; Guo et al., 2021; Wu et al., 2021) during the retraining. Specifically, we show win rates of the agents in *YouShallNotPassHumans*, *KickAndDefend*, *SumoHumans* and non-loss rate in *SumoAnts*. The lowest rate of each agent is depicted with a dashed line. More details are shown in Table 7 in Appendix A.7.

Table 1: The non-loss rates of our adversarial agent and baseline adversarial agents against masked victim agents in 100 games. Each experiment has been done 4 times. 'Before' and 'After' indicate before and after masking the victim agent. All data is reported as mean±std, rounded to the nearest integer.

| Environment | Ours (%) | | Gleave *et al.* (%) | | Guo *et al.* (%) | | Wu *et al.* (%) | |
|---|---|---|---|---|---|---|---|---|
| | Before | After | Before | After | Before | After | Before | After |
| YouShallNotPassHumans | 96 ± 2 | 73 ± 4 | 66 ± 2 | 0 ± 0 | 72 ± 3 | 0 ± 0 | 50 ± 2 | 0 ± 0 |
| KickAndDefend | 93 ± 4 | 7 ± 1 | 65 ± 1 | 3 ± 1 | 63 ± 2 | 5 ± 1 | 69 ± 4 | 5 ± 1 |
| SumoAnts | 83 ± 2 | 81 ± 2 | 76 ± 2 | 69 ± 4 | 81 ± 3 | 78 ± 1 | 57 ± 5 | 53 ± 4 |
| SumoHumans | 92 ± 1 | 90 ± 1 | 92 ± 1 | 91 ± 1 | 92 ± 0 | 91 ± 1 | 94 ± 2 | 92 ± 1 |

but also incorporates non-adversarial actions like obstructing the victim with its body to win the game. Based on this finding, we demonstrate that in *YouShallNotPassHumans*, our agent can defeat the victim by performing adversarial and non-adversarial actions simultaneously.

## 4.3 Comparison with Existing Episodic Control Method

Since episodic control shares a similar insight with our approach and serves as the basis for our method, to show the effectiveness of our method, we manually implement the state-of-the-art episodic control method NECSA (Li et al., 2023), integrating it with PPO to train the agents to attack the victims. To facilitate a fair comparison, we specifically adjust NECSA to also investigate how states contribute to winning or losing outcomes. NECSA uses a grid-based tabular episodic memory and groups similar states into the same grid, and then adds the average evaluation of that grid to the state's reward.

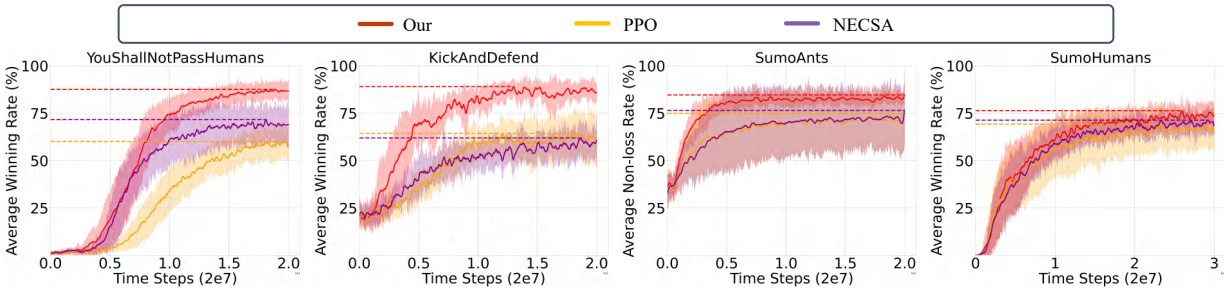

Figure 4: The comparison of performances between our method and NECSA. We show win rates of the agents in *YouShallNotPassHumans*, *KickAndDefend*, *SumoHumans* and non-loss rate in *SumoAnts*. The highest rate of each agent is depicted with a dashed line. More details are shown in Table 8 in Appendix A.7.

Figure 4 presents a performance comparison between our agents and the NECSA agents across four environments. The experimental results indicate that NECSA is less effective in all environments and even hinders the training process in the *KickAndDefend* environment. During NECSA's training, we observe that it repeatedly rewards states from losing episodes, which contradicts the fundamental principle of reward design in two-player competitive games. We believe this suboptimal reward engineering strategy diminishes NECSA's effectiveness, further underscoring the advantage of our conditional reward revision approach.

Table 2: The mean training time (rounded to the nearest hour) for a single run with PPO, our method, and NECSA.

| Environment | Methods | | |
|---|---|---|---|
| | PPO | Ours | NECSA |
| YouShallNotPassHumans | 10 | 12 | 13 |
| KickAndDefend | 22 | 24 | 26 |
| SumoAnts | 30 | 33 | 35 |
| SumoHumans | 24 | 26 | 29 |

Both our method and NECSA require additional time for state evaluation and reward engineering, but our approach leverages a neural-network-based memory to reduce this time cost. Table 2 compares training times for PPO, our method, and NECSA. We observe that our method's overhead is approximately two hours which is generally acceptable in practice, and is shorter than NECSA's training time across all environments.

## 4.4 Experiments in StarCraft II

To further evaluate whether our method is effective in a more complex two-player competitive environment, we conducted experiments in StarCraft II. We begin with two agents sharing the same initial policy: one serves as the fixed victim agent, and the other is an adversarial agent trained using our method and the vanilla PPO. As shown in Figure 5, the agent trained with our method achieves a 10% higher win rate than the agent trained with PPO, demonstrating that our approach is still effective in more sophisticated environments.

However, the performance gain is smaller than what we observed in traditional MuJoCo game environments. We suspect this is because StarCraft II has a much larger state vector, making it difficult for the LSTM network to encode states into suitable abstract representations. Replacing the LSTM with a more powerful architecture such as a Transformer may further enhance the effectiveness of our method in such complex environments.

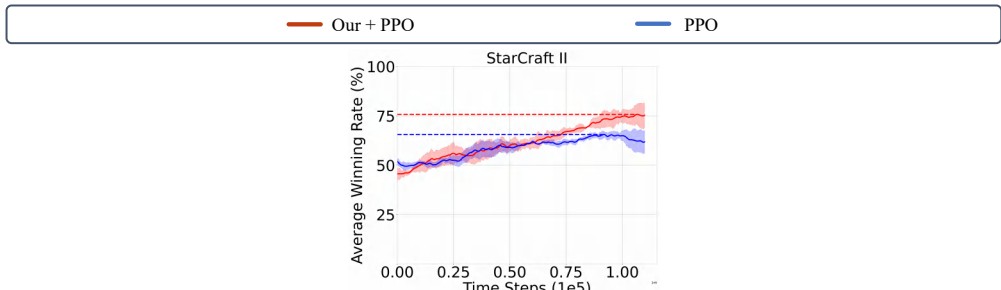

Figure 5: The comparison of the win rates between our method and PPO in StarCraft II.

## 4.5 Ablation Study

### 4.5.1 Pattern

As stated in Section 3.2.1, we evaluate patterns instead of states to generate the historical scores. The length of these patterns, however, introduces a critical trade-off. While longer patterns can provide richer

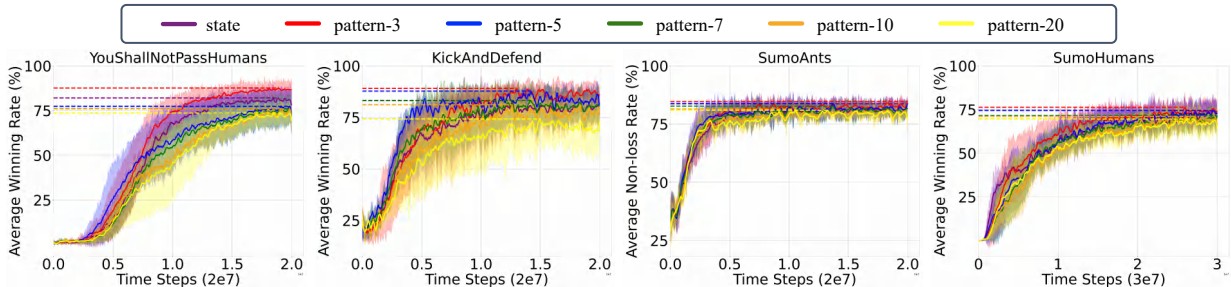

Figure 6: The comparison of performance between agents guided by historical evaluations of patterns with varying lengths. We show win rates of the agents in *YouShallNotPassHumans*, *KickAndDefend*, *SumoHumans* and non-loss rate in *SumoAnts*. The highest rate of each agent is depicted with a dashed line. More details are shown in Table 9 in Appendix A.7.

temporal context, they also risk suffering from the "curse of the pattern length", where an exponentially growing pattern space causes any specific long pattern to be observed so rarely that its historical score is based on too few related samples to be a reliable average, resulting in a noisy, high-variance learning signal.

To investigate this trade-off, we compare the performances of agents with different input pattern lengths. In Figure 6, we can see that performance initially improves when moving from single states (k=1) to short patterns (k=3 and k=5), but starts to degrade for longer patterns (k=7, 10, 20), especially in *YouShallNot-PassHumans* and *KickAndDefend*. This finding confirms both the benefit of using patterns and the critical importance of selecting an appropriate length (e.g., 3) to balance the trade-off.

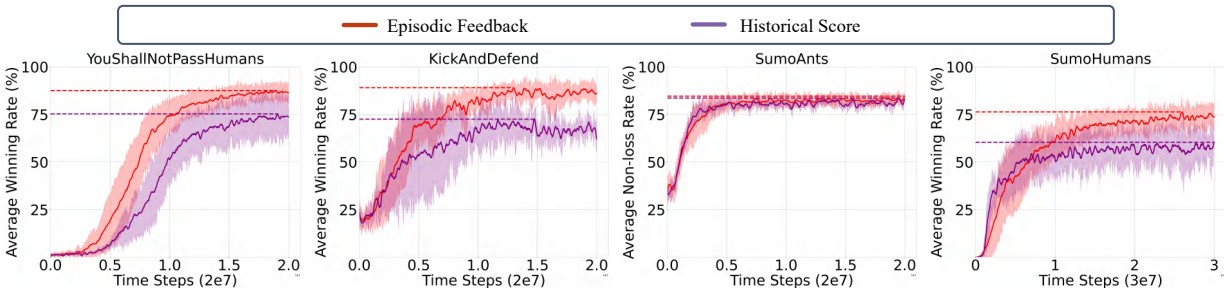

Figure 7: The comparison of performances between proposed training approach guided by episodic feedbacks and historical scores. We show win rates of the agents in *YouShallNotPassHumans*, *KickAndDefend*, *SumoHumans* and non-loss rate in *SumoAnts*. The highest rate of each agent is depicted with a dashed line. More details are shown in Table 10 in Appendix A.7.

### 4.5.2 Episodic Feedback

In Section 3.2.2, we mention that the episodic feedback of a pattern is computed by the difference between the historical score and the average cumulative reward of the group containing the episode. If episodic feedback is greater than 0, the pattern can be considered a better pattern than patterns in recently generated episodes, thereby enabling the adversarial agent to search for an optimal strategy. To show the effectiveness of episodic feedback, we use both episodic feedback and historical score to revise the reward. Based on the results shown in Figure 7, agents trained with episodic feedback achieve approximately 10% higher win rates than those trained with historical scores across YouShallNotPassHumans, KickAndDefend, and SumoHumans. This indicates that, compared to historical scores, using episodic feedback for reward revision enables the agent with the ability of discovering a better policy.

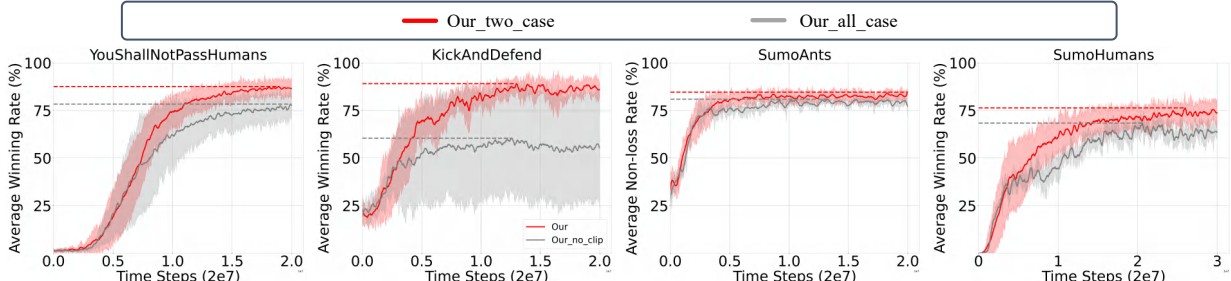

Figure 8: The comparison of performances between proposed training approach with different revision conditions. We show win rates of the agents in *YouShallNotPassHumans*, *KickAndDefend*, *SumoHumans* and non-loss rate in *SumoAnts*. The highest rate of each agent is depicted with a dashed line. More details are shown in Table 11 in Appendix A.7.

### 4.5.3   Revision Condition

As noted in Section 3.2.2, we do not revise rewards in certain scenarios, such as when a state from a winning episode for the adversarial agent receives negative episodic feedback. If we were to extend reward revision to these scenarios, some states leading to losses would be rewarded, while some states leading to wins would be penalized, potentially causing the adversarial agent to learn a losing strategy. To demonstrate this effect, we also applied reward revisions in these additional cases and present the results in Figure 8. We observe that while the peak performance of the "our_two_case" agent is only about 3% higher than that of the "our_all_case" agents, the average performance degrades substantially under full reward revision, especially in *KickAndDefend*. Similar to the results of NECSA stated in Section 4.3, this drop likely arises because the suboptimal reward engineering hinders the agent's ability to learn a winning strategy, resulting in unstable learning outcomes. In fact, we also find that states in losing episodes received positive rewards and states in winning episodes received negative rewards in all four environments when we do full reward revision. This finding further shows that reward engineering must adhere to the rules of two-player competitive games.

## 5   Discussion

As stated in Section 4.1, our method utilizes the PPO algorithm as the fundamental training method to train the adversarial agents. Additionally, our method can also be applied to various DRL algorithms. In the Appendix, we demonstrate the performances of our approach applied to the baseline algorithms (Wu et al., 2021; Guo et al., 2021) and compare the results with those of the original baseline algorithms in *YouShallNotPassHumans* and *KickAndDefend*. The results show that by leveraging historical evaluations to revise the rewards, the performances of all baseline approaches get improved.

In Section 3, we primarily explain episodic feedback as a reward-shaping function. It is also important to note that incorporating episodic feedback as a shaping term is functionally similar to the use of a state-dependent baseline (Williams, 1992) in policy-gradient methods, given the nature of the historical score. This score, calculated as the average cumulative return over past episodes that include a specific state (pattern), can be viewed as a Monte Carlo value estimate with a discount factor of 1 under a sparse-reward setting in which the reward equals the total episode return. This motivates an analogy with a state-dependent baseline. However, the application mechanism differs: a standard baseline only changes the gradient calculation term, whereas our historical score serves as a reward shaping term that directly modifies the step rewards. While modifying rewards can bias the policy gradient, our episodic feedback ensures stability. As proven in Appendix A.2, the episodic feedback, with zero mean given the state, preserves the expected policy gradient, thereby leaving the optimal policy unchanged.

While the functional similarity to potential-based reward shaping and state-dependent baselines provides theoretical context, it does not fully explain the substantial performance improvements observed in our experiments. The challenge for a complete theoretical analysis centers on the issue of reward variance:

specifically, how changes in step rewards using episodic feedback affect the learning process, even when all other aspects (e.g., states, actions, optimal policy, expected policy gradient) of the training algorithm remain the same. This issue is not solved in our paper, and we think a further study of it would be both difficult and highly valuable for future research.

## 6 Conclusion

In this paper, we propose an adversarial policy training method with reward engineering to train an adversarial agent more effectively and efficiently. Our method introduces an episodic memory to utilize the historical experiences to generate historical evaluations for the states and consequently revise the rewards of the states based on the evaluation, thereby integrating the historical evaluations into the rewards used for adversarial training to emphasize the high-performing states. In our experiments, we demonstrate that agents trained with our approach achieve the most promising attack performance and defense difficulty. Additionally, by comparing the behaviors of adversarial agents, we discover that our attack method can explore different but optimal strategies by integrating successful actions from historical experiences. We believe our exploration of game states and use of historical experiences to improve the rewards advance adversarial policy training methods.

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

# A  Appendix

## A.1  Policy–optimality guarantee of episodic feedback

Consider an MDP $M = \langle S, A, T, \gamma, R \rangle$ and the shaped MDP $M' = \langle S, A, T, \gamma, R' \rangle$ with

$$R'(s, a, s') \;=\; R(s, a, s') \;+\; \delta(s),$$

where $\delta : S \to \mathbb{R}$ is the *episodic-feedback*.

For any policy $\pi$ and initial state–action pair $(s, a)$,

$$
\begin{aligned}
Q'_\pi(s, a) &= \mathbb{E}_\pi\left[\sum_{t=0}^{\infty} \gamma^t R'(s_t, a_t, s_{t+1})\right] \\
&= \mathbb{E}_\pi\left[\sum_{t=0}^{\infty} \gamma^t \big(R(s_t, a_t, s_{t+1}) + \delta(s_t)\big)\right] \\
&= Q_\pi(s, a) \;+\; \underbrace{\mathbb{E}_\pi\left[\sum_{t=0}^{\infty} \gamma^t \delta(s_t)\right]}_{(*)}.
\end{aligned}
\tag{6}
$$

As stated in Equation 2, episodic feedback is defined as

$$\delta(s) \;=\; h\_score(s) \;-\; \overline{\mathcal{R}},$$

where $h\_score(s)$ is the empirical mean[1] of the episode return, and $\overline{\mathcal{R}}$ is the running mean of the average cumulative reward over *all* past episodes. Both quantities are unbiased estimates of the same random variable, the episode cumulative reward $R_{\text{cum}}$. Hence

$$\mathbb{E}_\pi\big[h\_score(s_t)\big] \;=\; \mathbb{E}_\pi\big[\overline{\mathcal{R}}\big] \;=\; \mathbb{E}_\pi[R_{\text{cum}}]. \tag{7}$$

Equation 7 immediately gives $\mathbb{E}_\pi[\delta(s_t)] = 0$; substituting into $(*)$,

$$\mathbb{E}_\pi\left[\sum_{t=0}^{\infty}\gamma^t\delta(s_t)\right] \;=\; \sum_{t=0}^{\infty}\gamma^t\,\mathbb{E}_\pi[\delta(s_t)] \;=\; 0.$$

Because the correction term is zero, Equation 6 reduces to $Q'_\pi(s,a) = Q_\pi(s,a)$ for every $(s,a)$ and every policy $\pi$. The optimal-action sets therefore coincide:

$$\arg\max_a Q'_\pi(s,a) \;=\; \arg\max_a Q_\pi(s,a), \quad \forall s.$$

Hence every optimal policy in $M'$ is also optimal in $M$. In other words, the episodic feedback $\delta(s)$ acts like a potential-based shaping function (Ng et al., 1999), which shifts all $Q_\pi(s,\cdot)$ values by the same constant and thus leaves the policy optimality unchanged.

## A.2 Policy-gradient Preservation

Let the shaped reward be $r'_t = r_t + \delta_t$. Define discounted returns

$$G_t \;\coloneqq\; \sum_{l=0}^{\infty}\gamma^l r_{t+l}, \quad G'_t \;\coloneqq\; \sum_{l=0}^{\infty}\gamma^l r'_{t+l} = \sum_{l=0}^{\infty}\gamma^l(r_{t+l} + \delta_{t+l}) = G_t + B_t, \quad B_t \;\coloneqq\; \sum_{l=0}^{\infty}\gamma^l\delta_{t+l}. \tag{8}$$

For policy gradient $\nabla_\theta J(\theta)$ and revised policy gradient $\nabla_\theta J'(\theta)$, subtracting yields

$$\nabla_\theta J'(\theta) - \nabla_\theta J(\theta) = \mathbb{E}\Big[\sum_{t=0}^{\infty}\nabla_\theta \log \pi_\theta(a_t \mid s_t)\,G'_t\Big] - \mathbb{E}\Big[\sum_{t=0}^{\infty}\nabla_\theta \log \pi_\theta(a_t \mid s_t)\,G_t\Big]$$

$$= \mathbb{E}\Big[\sum_{t=0}^{\infty}\nabla_\theta \log \pi_\theta(a_t \mid s_t)\,B_t\Big] \tag{9}$$

According to Equation 7 in Appendix A.1, $\mathbb{E}[B_t \mid s_t] = \mathbb{E}[\delta(s_t)] = 0$,

$$\mathbb{E}[\nabla_\theta \log \pi_\theta(a_t \mid s_t)\,B_t] = 0,$$

therefore, $\nabla_\theta J'(\theta) = \nabla_\theta J(\theta)$ and the policy gradient is unbiased.

## A.3 Hyper-parameter Analysis

For PPO hyper-parameter selections, we use the same parameters from Gleave et al. (2020). The hyper-parameters for our episodic memory are listed in Table 3. As mentioned in Li et al. (2023), 0.1 for $\epsilon$ works well for episodic control methods in MuJuCo games, so we follow this setting in our experiments. We further analyze the rest two hyper-parameters.

### A.3.1 Pattern Length

As stated in Section 3.2, we use state sequences, referred to as patterns, as the basis for performance evaluation in past episodes. To find out the best parameter for the length of patterns, we conduct experiments with different pattern lengths and show the results in Figure 9. We have selected six different pattern lengths in our experiments. We can see from Figure 9 that the agents reach the best performances when the pattern length is 3.

---

[1] In practice $h\_score(s)$ is the running average of the undiscounted cumulative reward over *all past episodes that contained $s$*.

Table 3: Hyper-parameters of episodic memory used in our experiments.

| Hyper-parameter | |
| --- | --- |
| Pattern Length k | 3 |
| Group Size n | 100 |
| Epsilon $\epsilon$ | 0.1 |

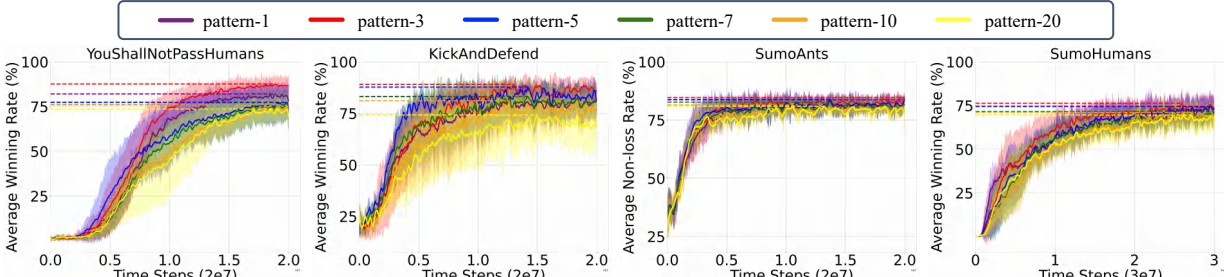

Figure 9: The performance comparison between our adversarial agent with different input pattern lengths.

### A.3.2 Group Size

In Section 3.2.2, we compare the historical score of a pattern with an average cumulative reward of recently generated past episodes to calculate episodic feedback. Group size $n$ is introduced to control the number of past episodes. If the group size is too large, some states that lead to bad patterns may be erroneously rewarded, and the magnitude of rewards and penalties is reduced, thereby weakening the ability to find the optimal policy. On the other hand, if the group size is too small, it is more likely to wrongly penalize good states and reward bad states. Therefore, we conduct an analysis of 5 group sizes which are *50*, *100*, *150*, *200* and *300*. With the result shown in Figure 10, agents generally achieve better performance with a group size of 100. However, the unstable results in *SumoHumans* environment highlight the importance of carefully selecting this parameter.

### A.4 Episodic Memory

In this section we share more details about the implementation of the episodic memory introduced in Section 3.3.1.

Table 4: The architecture of the episodic memory

| Module | shape |
| --- | --- |
| LSTM | (64, 256, 1) |
| linear1 | (256, 512) |
| Tanh | |
| linear2 | (512, 1) |

In Table 4, we give the architecture of the episodic memory. The episodic memory consists of a LSTM and a MLP (linear1, Tanh, linear2). 64 in the shape of LSTM refers to the length of the state from environment and 256 refers to the hidden length of LSTM. The LSTM is used to encode a pattern into an abstract vector so that the MLP can process. The MLP is used to output the historical score of the pattern, which evaluates the average performance of the pattern in past episodes.

In the Algorithm 2, we show the forward process of the episodic memory. The LSTM receives a pattern $p$ as input and outputs an output sequence, the last hidden state vector and the last cell state vector of LSTM. The last hidden state vector will be processed with MLP, under the order of linear1, Tanh, linear2 and the MLP will output the historical score $h\_score$ of the input pattern.

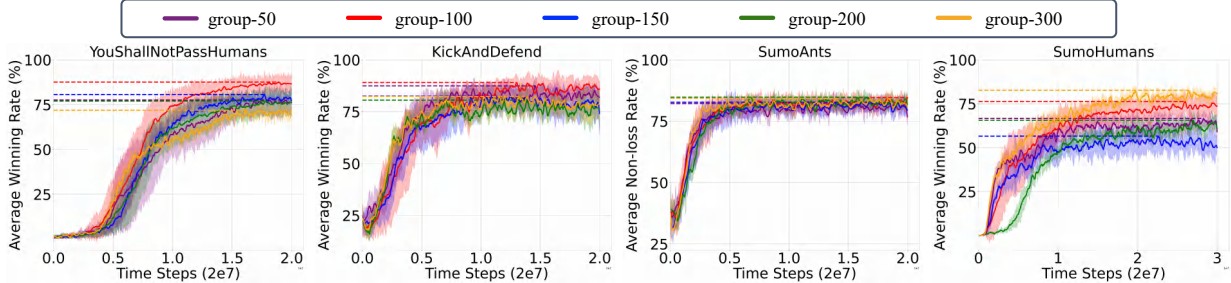

Figure 10: The comparison of win rate between our adversarial agent with different group size.

---

**Algorithm 2** Forward process of the episodic memory.

---

**Input**: pattern $p$ (shape:[3, 64])
**Output**: historical score $h\_score$ (shape:1)

1: output, hidden, cell = LSTM($p$)
2: $h\_score$ = linear1(hidden[-1,])
3: $h\_score$ = Tanh($h\_score$)
4: $h\_score$ = linear2($h\_score$)

---

In the Algorithm 3, we show the update process of the episodic memory. After one episode is ended, we extract patterns from the episode and calculate the cumulative reward of the episode. Then, we predict the historical score for each pattern and calculate the MSELoss between the historical score and the cumulative reward. The loss will be backpropagated to update the network.

### A.5 Generalizability Analysis

In Section 5, we state that our approach can be applied to various DRL algorithms. Since Gleave *et al.* can be seen as PPO, we adopt our approach on the other two baseline attacks (Guo et al., 2021; Wu et al., 2021) and compare the performances with them. The results are shown in Figure 11. We can see the baselines adopting our episodic memory outperform the original baselines in *YouShallNotPassHumans* and *KickAndDefend*.

### A.6 Videos Of Experiments

Due to the limited maximum file size for the supplementary materials, we have uploaded the videos mentioned in Section 4.2 at `https://drive.google.com/drive/folders/1lJmWA7y8-1nMs_kOwzGlIMkjkPh2QVF6?usp=drive_link`.

### A.7 Main Results Supplementary

Supplementary tables of the figures in the main text are provided on the subsequent pages.

Table 5: The highest win rates of our agents and agents attacks against zoo victim agents are shown in Figure 2(a).

| Environment | Our (%) | Gleave *et al.* (%) | Guo *et al.* (%) | Wu *et al.* (%) |
|---|---|---|---|---|
| YouShallNotPassHumans | $87.62 \pm 7.38$ | $60.08 \pm 6.22$ | $65.77 \pm 7.60$ | $48.60 \pm 8.99$ |
| KickAndDefend | $89.06 \pm 7.98$ | $64.37 \pm 8.61$ | $65.32 \pm 6.92$ | $64.76 \pm 8.55$ |
| SumoAnts | $5.18 \pm 1.27$ | $5.19 \pm 2.10$ | $4.70 \pm 1.43$ | $8.14 \pm 2.87$ |
| SumoHumans | $76.35 \pm 8.29$ | $69.24 \pm 12.16$ | $64.49 \pm 7.15$ | $62.22 \pm 16.86$ |

---

**Algorithm 3** Update process of the episodic memory.

---

**Input**: episode $e$

1:  $P = \text{Pattern}(e)$
2:  $R = \text{Cumulative\_Reward}(e)$
3:  **for** $p$ in $P$ **do**
4:    $h\_score = \text{Memory}(p)$
5:    $\text{loss} = \text{MSELoss}(h\_score, R)$
6:    $\text{loss.backpropagation}()$
7:  **end for**

---

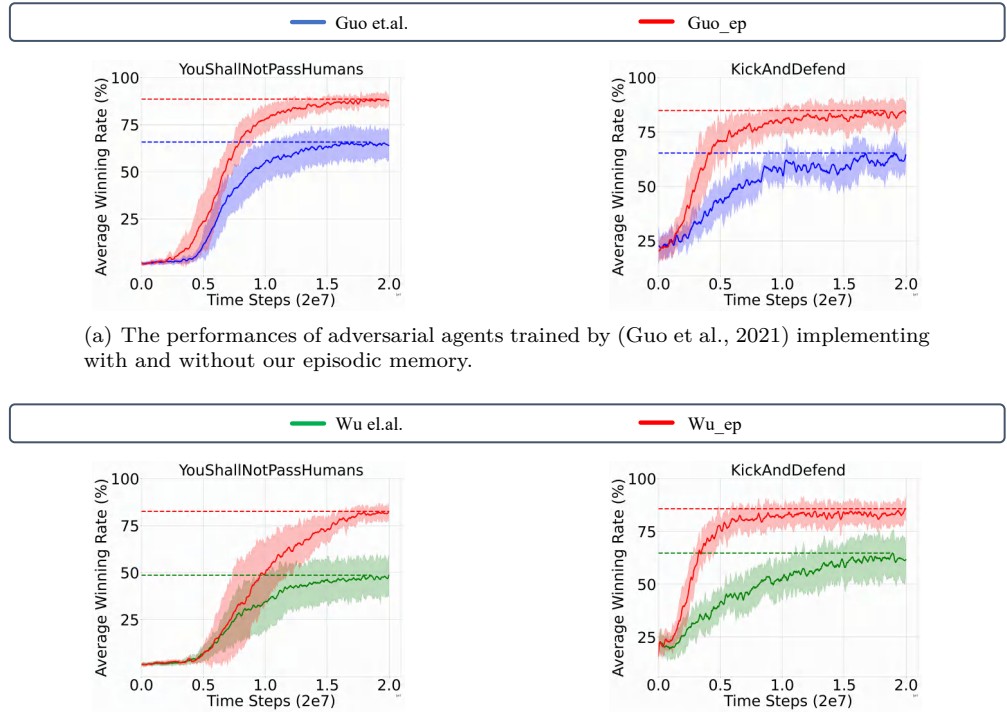

(a) The performances of adversarial agents trained by (Guo et al., 2021) implementing with and without our episodic memory.

(b) The performances of adversarial agents trained by (Wu et al., 2021) implementing with and without our episodic memory.

Figure 11: The performance comparison of win rate between agents trained by (Wu et al., 2021; Guo et al., 2021) implementing with and without our episodic memory in *YouShallNotPassHumans* and *KickAndDefend*.

Table 6: The highest non-loss rates of our agents and baseline agents against zoo victim agents are shown in Figure 2(b).

| Environment | Our (%) | Gleave *et al.* (%) | Guo *et al.* (%) | Wu *et al.* (%) |
|---|---|---|---|---|
| YouShallNotPassHumans | $87.62 \pm 7.38$ | $60.08 \pm 6.22$ | $65.77 \pm 7.60$ | $48.60 \pm 8.99$ |
| KickAndDefend | $90.01 \pm 7.56$ | $65.17 \pm 8.87$ | $66.56 \pm 7.14$ | $65.27 \pm 8.45$ |
| SumoAnts | $84.66 \pm 3.92$ | $74.94 \pm 16.24$ | $82.98 \pm 3.73$ | $40.97 \pm 6.65$ |
| SumoHumans | $91.68 \pm 7.52$ | $91.88 \pm 12.18$ | $90.49 \pm 5.48$ | $92.55 \pm 14.40$ |

Table 7: The performances of our agents and baseline agents against retrained victim agents are shown in Figure 3. We show win rates of the agents in *YouShallNotPassHumans*, *KickAndDefend*, *SumoHumans* and non-loss rate in *SumoAnts*.

| Environment | Our (%) | Gleave *et al.* (%) | Guo *et al.* (%) | Wu *et al.* (%) |
|---|---|---|---|---|
| YouShallNotPassHumans | $50.27 \pm 13.03$ | $5.00 \pm 2.50$ | $6.22 \pm 2.82$ | $5.99 \pm 2.99$ |
| KickAndDefend | $51.82 \pm 6.84$ | $28.02 \pm 7.33$ | $29.33 \pm 9.84$ | $32.38 \pm 9.54$ |
| SumoAnts | $83.15 \pm 2.98$ | $79.78 \pm 2.28$ | $82.49 \pm 2.77$ | $90.13 \pm 3.42$ |
| SumoHumans | $6.17 \pm 7.60$ | $6.03 \pm 5.35$ | $7.61 \pm 5.88$ | $10.71 \pm 6.95$ |

Table 8: The comparison of performances between our method and NECSA. The performances of our agents and NECSA against victim agents are shown in Figure 3. We show win rates of the agents in *YouShallNotPassHumans*, *KickAndDefend*, *SumoHumans* and non-loss rate in *SumoAnts*.

| Environment | PPO (%) | Our (%) | NECSA (%) |
|---|---|---|---|
| YouShallNotPassHumans | $60.08 \pm 6.22$ | $87.62 \pm 7.38$ | $71.71 \pm 9.18$ |
| KickAndDefend | $74.94 \pm 16.24$ | $89.06 \pm 7.98$ | $76.52 \pm 16.31$ |
| SumoAnts | $83.15 \pm 2.98$ | $84.66 \pm 3.92$ | $82.49 \pm 2.77$ |
| SumoHumans | $69.24 \pm 12.15$ | $76.34 \pm 8.29$ | $71.34 \pm 7.38$ |

Table 9: The performances of our agents guided by historical evaluation of patterns with different lengths against zoo victim agents is shown in Figure 6. We show win rates of the agents in *YouShallNotPassHumans*, *KickAndDefend*, *SumoHumans* and non-loss rate in *SumoAnts*.

| Environment | state (%) | pattern-3 (%) | pattern-5 (%) | pattern-7 (%) | pattern-10 (%) | pattern-20 (%) |
|---|---|---|---|---|---|---|
| YouShallNotPassHumans | $82.06 \pm 6.77$ | $87.62 \pm 7.38$ | $77.44 \pm 11.81$ | $76.22 \pm 7.26$ | $76.02 \pm 5.42$ | $73.66 \pm 9.61$ |
| KickAndDefend | $83.15 \pm 11.29$ | $89.06 \pm 7.98$ | $87.78 \pm 4.07$ | $83.24 \pm 9.10$ | $81.12 \pm 8.66$ | $76.84 \pm 9.12$ |
| SumoAnts | $83.73 \pm 4.15$ | $84.66 \pm 3.92$ | $83.83 \pm 4.15$ | $82.71 \pm 4.23$ | $81.47 \pm 3.18$ | $81.01 \pm 2.49$ |
| SumoHumans | $74.35 \pm 6.71$ | $76.35 \pm 8.29$ | $74.47 \pm 9.66$ | $71.56 \pm 8.45$ | $71.07 \pm 9.38$ | $69.60 \pm 7.14$ |

Table 10: The performances of our agents trained with episodic feedback and historical score against zoo victim agents shown in Figure 7. We show win rates of the agents in *YouShallNotPassHumans*, *KickAndDefend*, *SumoHumans* and non-loss rate in *SumoAnts*.

| Environment | Episodic Feedback (%) | historical score (%) |
|---|---|---|
| YouShallNotPassHumans | $87.62 \pm 7.38$ | $75.28 \pm 8.25$ |
| KickAndDefend | $89.06 \pm 7.98$ | $72.85 \pm 11.48$ |
| SumoAnts | $84.66 \pm 3.92$ | $83.66 \pm 2.30$ |
| SumoHumans | $76.35 \pm 8.29$ | $72.15 \pm 6.90$ |

Table 11: The performances of our agents with and without revision conditions against zoo victim agents is shown in Figure 8. We show win rates of the agents in *YouShallNotPassHumans*, *KickAndDefend*, *SumoHumans* and non-loss rate in *SumoAnts*.

| Environment | Our_two_case (%) | Our_all_case (%) |
|---|---|---|
| YouShallNotPassHumans | $87.62 \pm 7.38$ | $78.39 \pm 10.74$ |
| KickAndDefend | $89.06 \pm 7.98$ | $60.45 \pm 25.15$ |
| SumoAnts | $84.66 \pm 3.92$ | $80.44 \pm 3.73$ |
| SumoHumans | $76.35 \pm 8.29$ | $68.40 \pm 4.14$ |

