# OpenReview forum: "Improving Adversarial Training for Two-player Competitive Games via Episodic Reward Engineering"
_TMLR — Accepted by TMLR_

### Review · Reviewer_AeY8 · 2025-08-25

**Summary Of Contributions:**

The authors present a novel approach in the area of on adversarial policy training in two-player scenarios that enhances the existing PPO algorithm with a neural episodic memory to provide more fine-grained feedback on sequences within observed episodes. The introduced memory acts as an additional component that allows to detect desired positive or punish undesired negative 'patterns' (subsequences of an episode of fixed length that are indicators to a win or loss) that have been encountered during previous runs. The episodic memory is trained alongside the adversarial agent and the authors claim a better performance is comparison to existing baseline methods in the area is achieved.  The authors introduce and test a 'conditional reward revision', which limits memory guidance  patterns that rank above (or below) average particular groups of episodes (to account for later episodes consistently achieving higher rewards earlier patterns). The method and ablations are tested on four Mujoco two-player game environments, as well as on a more complex StarCraft II environment, showing good performance across all evaluations and against ablations.



**Strengths**

1) The idea of leveraging a neural episodic memory is intuitive. The paper is overall well written and the approach is presented in simple successive steps. Figures and plots are easy to read and interpret.
2) The presented formalism follows common notation and is formalized to a sufficient degree. The presented proof of policy-optimality under episodic feedback seems to hold.
3) The overall selection of environments and experimental setup seems reasonable and follows existing baseline experiments. Experiments reveal strong improvements in two of the four Mujoco environments and the authors are able to present a 10% score increase for a more complex StarCraft II scenario. The authors compare to several baselines in the field of adversarial RL and a closely related NECSA baseline with episodic control. They present several ablations with regard to the individual components of their approach and also inspect countermeasures of 'victim retraining' and 'masked observation defense' against their approach.



**Weaknesses**

1) While PPO is a still widely considered methods and given the strong claims on performance improvements, the paper focuses strongly on methods within the subfield, but it (or possibly the general field of adversarial attacks in RL) seems to lack behind general advances in RL (e.g., [1-3]). It is unclear to me, whether similar advantages could be achieved in the presence of possibly stronger algorithms.
2) Variation experiments on pattern length (A.2.1) and group size (A.2.2) only consider two of the four environments in the main paper (admittedly one of the remaining four environments shows low winning performance) and evaluate only two alternative values for pattern length and group size. While the alternative variants demonstrate a parameter choice above and below the optimal chosen value, it is hard to assess the fidelity of parameters from only two alternative choices on two environments.
3) The general setup described of Sec. 4.4.1 is unclear to me: How is the mentioned state guidance implemented? Does this imply a 1-step pattern - or is the episodic feedback omitted in this case? The authors furthermore mention "historical scores of both states" and to "[...] revise the rewards with two episodic feedbacks". I do not understand which "both states" are meant here and how (and why) two feedbacks are computed.
4) Algorithm 1, line 9, states "if The episode is the ith episode in Group m then ...". I am wondering under which condition this statement will evaluate to false. To my understanding an episode should included in some group (with some index i assigned to it), such that it could be written as a normal assignment. Maybe I misunderstood something here and would like to ask the authors to clarify.



[1] Badia, Adrià Puigdomènech, et al. "Agent57: Outperforming the atari human benchmark." *International conference on machine learning*. PMLR, 2020.
[2] Hessel, Matteo, et al. "Muesli: Combining improvements in policy optimization." *International conference on machine learning*. PMLR, 2021.
[3] Kapturowski, Steven, et al. "Human-level Atari 200x faster." *arXiv preprint arXiv:2209.07550* (2022).

**Audience:**

Yes

**Audience Explanation:**

The introduction of a neural episodic memory in the context of adversarial learning is an interesting idea that seems to show clear improvement over existing methods and might spark further interest in the general application of adversarial methods in RL. The presented experiments and comparison to a recent baseline with episodic memory confirm a robust performance of the approach and might be suited to be considered as an additional component to more recent algorithms.

**Broader Impact Concerns:**

None.

**Claims And Evidence:**

Yes

**Claims Explanation:**

While I am not intimately familiar with existing literature in this particular subfield, the presented experiments seem to clearly demonstrate improvements over existing algorithms in the field of adversarial RL and show improvements under the more complex StarCraft II setting. My main concern are the lack and comparisons to more recent RL methods under which the presented advantages might vanish (Weakness 1).

Similarly, the presented variations and ablations seem reasonable and support the necessity of individual components and choice of parameters. The sparse variations of parameters in combination with evaluation on only two environments limit insights with regard to the choice of pattern length and group size (Weakness 2).

**Requested Changes:**

The critical requested changes mainly regard to mentioned weaknesses:

1) Given the strong claims on the performance improvements of the methods, I consider weakness 1 on comparisons/integration of the idea with more recent methods to be most relevant and critical. Can the authors confirm, that the observed performance improvements still persist under more recent, possibly stronger, algorithms? Since I'm not well versed in literature of the particular subfield, I might have missed a key point that prevents more recent methods from being applied. In this case I would like to ask the authors to elaborate on why the application of such algorithms might not make a difference or possibly argue why they are out of scope for this paper.
2) Similarly, I would like to ask the authors to clarify weaknesses 3 and 4 mentioned above. While I consider these to be critical, I believe that these points arose mainly due to a lack of explanation or confusion about the exact setup of the method.



**Non-critical**

1) I believe that the paper might be strengthened by probing further values of pattern length and group size, (possibly also testing on the remaining environments) to get a better perspective on the importance of parameter choice. However, since the method seems to be working without strong fine-tuning of the parameters, I consider this point to be non-critical.
2) [Minor] While Figure 1 showcases the whole workflow of the method, it is, at first, rather unclear which parts constitute the novel contributions of the paper in contrast to existing frameworks. The authors might want to highlighting the respective components in the figure and/or state the novel parts more explicitly in the the caption.



**Typos**

* Sec. 3, "Our threat model: [...] and assume[s] that"
* Sec. 4.1, "For the hyper-parameters, we set the [epsilon?] as 0.1".
* Sec. 4.4.3, last line p. 11: "In fact, [w]e ..."
* Sec. 3.3.2, "the first i episode" -> "the first i episode[s]"

---

> ### Author Response · Authors · 2025-08-29
> **Response to Reviewer AeY8**
>
> We appreciate the time and effort you've dedicated to reviewing our work and your invaluable and enlightening review. We will fix the typos and carefully revise our manuscript based on your reviews.
>
> ---
>
> Weakness 1:
>
> Agent57 is an Atari-targeted, single-agent algorithm that combines R2D2 with NGU. To the best of our knowledge, there are no prior two-player competitive RL papers (including self-play and adversarial training) that adopt Agent57 or Agent57-derived stacks as the base learner, and we are not aware of any official implementation of Agent57 that can be transplanted to our setting. Importantly, the current implementation of R2D2 supports only discrete action spaces [1], which matches Atari but does not match our MuJoCo competitive tasks, where actions are continuous. Our experiments follow the standard competitive self-play [2] setup; implementing an Agent57 for this MuJoCo setup needs a reconstruction of its structure, which may amount to a different algorithm.
>
> Although we cannot directly run Agent57-based methods in our experiment environments, we believe that our approach is compatible with stronger base learners. Similar to our method, NGU is also an episodic control-based method. Different from their intrinsic reward combining episodic novelty (pseudo step counts) and life-long novelty scaling (RND), our method uses episodic feedback which learns how states or patterns relate to the eventual game outcome (win vs. loss) across episodes. Importantly, the game environment does not supply per-step outcome information, so our revision provides a dense, outcome-aligned learning signal to the rewards. As shown in our experiments, our use of such learning signals is able to vastly improve PPO. Since NGU/Agent57 were not designed to capture and explore such outcome information, our method should be able to improve these methods under two-player competitive game environments. In the appendix, we also report results applying our reward revision to stronger PPO variants (e.g., Guo et al., Wu et al.) in the field and observe consistent improvements, indicating that the gains are not tied to vanilla PPO.
>
> [1] https://di-engine-docs.readthedocs.io/en/latest/12_policies/r2d2.html
>
> [2] Trapit Bansal, Jakub Pachocki, Szymon Sidor, Ilya Sutskever, and Igor Mordatch. Emergent complexity via multi-agent competition. In International Conference on Learning
>
> ---
>
> Weakness 2:
>
> Although we did not retain the logs, we have done experiments with pattern lengths 7, 10, and 20 in YouShallNotPassHumans and KickAndDefend. In both tasks the benefit levels off after length-5, and in YouShallNotPassHumans it drops sharply at lengths 10 and 20 (while still remaining effective compared to the baselines). We believe this decline relates to the size and composition of the environment’s state vector. We can provide further analysis of more parameter selections in our revised script.
>
> ---
>
> Weakness 3:
>
> This is the sentence: ‘we calculate the episodic feedback with historical scores of both states and 3-step patterns and then revise the rewards with two episodic feedbacks.’ The two episodic feedback is for states (1-step patterns) and 3-step patterns. We will modify the sentence to ‘We compute episodic feedback from two sources: **states (1-step patterns)** and **3-step patterns**, and revise rewards using both signals.’ for better understanding.
>
> ---
>
> Weakness 4:
>
> Thanks for pointing out this mistake. Your understanding is right, this statement will always be satisfied so we should not use if but a normal assignment.
>
> ---
>
> We hope our responses and clarifications have addressed your concerns. If there are any remaining questions, please let us know, and we will do our best to resolve them.

---

> > ### Comment · Reviewer_AeY8 · 2025-09-02
> >
> > Dear authors,
> > thank you for your reply and your promise to improve on the writing and correcting the algorithm. I also appreciate the clarifications regarding Agent57.
> >
> > However, my concerns of proper variation experiments still remain. You mention that the "benefit levels off" for longer patterns on YouShallNotPassHumans and KickAndDefend. This might be a valuable insight, for which I would have expected empirical validation presented in the paper. Analogous to the comment of reviewer FGt1, pattern length seems to be one of the important hyper-parameters of the method and might strongly affect the horizon of the reward term, and overall convergence/learning quality. Similar considerations yield for the group size parameter. Overall, I would like to recommend to validate the robustness of the approach with respect to its hyper-parameters through a more thorough empirical analyses.
> >
> > Since I could not find any information on this while skimming through the paper: It would be helpful if the authors could report the used used hardware and runtimes, which would enable one to estimate the time and costs required of running algorithm (including the feasibility of requesting possible additional variations as the ones mentioned above.)

---

> > > ### Author Response · Authors · 2025-09-04
> > >
> > > Dear Reviewer AeY8
> > >
> > > Thank you for your response. We will add our hardware details to the experimental setup and the time cost is already reported in Table 2 from Section 4.3.
> > >
> > > Since we did not retain logs from our early experiments, we are rerunning them now. Based on our current estimate, we will be able to provide
> > > - a pattern-length ablation with more parameter selections on all four environments, and
> > > - a group-size ablation with more parameter selections on YouShallNotPassHumans and KickAndDefend.
> > >
> > > The group-size experiments on the other two environments may not finish before the discussion period ends. We plan to upload a revised manuscript once above mentioned experiments are completed.
> > >
> > > Thanks again for your helpful suggestions.

---

> > > > ### Comment · Reviewer_AeY8 · 2025-09-20
> > > >
> > > > Dear authors,
> > > > thank you for adding the various changes to the paper. Just a small comment: The pattern length ablation in figure 9 currently only shows the plots for pattern lengths 1 and 3, instead of all 6 variations (while being correctly described in the text and figure legend).

---

> > > > > ### Author Response · Authors · 2025-09-26
> > > > >
> > > > > Dear Reviewer AeY8
> > > > >
> > > > > Thanks for your response again. We have corrected the figures related to pattern length. Furthermore, we have included the results analyzing different group size in all four environments in the latest version.

---

### Review · Reviewer_FGt1 · 2025-08-28

**Summary Of Contributions:**

The paper aims to address issues with training an agent via deep reinforcement learning to attack another agent. The authors claim that fixing the agent being attacked and casting this as a standard RL problem can be enhanced. This is done via fitting zero-mean state-specific baselines via scoring trajectories with their average cumulative reward and demeaning with the average reward of similar episodes, and then injecting this into the environment reward to shape it.

## Strengths
1. The method is practical, and seems to perform well empirically.
2. The empirical evaluation is thorough and appears sound.

## Weaknesses
The weaknesses mostly pertain to what Action Editor ED9N pointed out. It does not seem like these were effectively addressed.
1. The authors do not thoroughly explore what their method means, how it is derived, and its relation to existing literature.
- The relation to a baseline is not explored, and the relation to a critic is hand-waved away.
- The relation to episodic control amounts to grouping by trajectories v.s. grouping by states, and so is a non-Markovian approach. Why exactly this produces any benefit in what is presumably an MDP is not explored.
2. The theory is shallow.

**Additional Comments:**

The claims made are largely supported by clear and convincing evidence, and some people would be interested in this work. Still, the weaknesses raised for rejection at the last submission were not effectively addressed, and so still need to be addressed now.

**Audience:**

Yes

**Audience Explanation:**

This relates to training deep RL agents to attack other agents when the other agent is fixed, which is an important enough application that there is enough literature on. The tweak appears to help empirically, for the above reasons stated in the last cell.

**Broader Impact Concerns:**

None.

**Claims And Evidence:**

Yes

**Claims Explanation:**

## Strengths

Regarding the main contributions stated the authors:
1. A novel policy learning framework with reward enhancement. This is true, the approach is evaluating similar trajectories instead of states is highlighted and explained, though the explanation could be cleared.
2. The second claim amounts to what their method is, which is correct.
3. The experiments are reasonably convincing, satisfying the third. Importantly, the method is viewed and developed from a practical lens, and so is reasonably easy to implement and has reasonable training time. An ablation study, as well as experiments on different environments against suitable baselines, are  provided.

## Weaknesses
1. **Relation to a critic, baseline, episodic control, etc.**
- **On baselines and critics.**
    - It is all well and good to cast this as a reward-shaping term, but this is quite different from vanilla reward-shaping. As Action Editor ED9N quite rightly points out, this is suspiciously similar to a critic. I would clarify that this looks closer to a baseline as in REINFORCE with baseline, instead of a TD(0) actor-critic.
    - This differs from a baseline in three ways. Firstly, the baseline is trajectory-specific, not state-specific. Secondly, the baseline is injected into each reward of the future trajectory, and not just the reward-to-go of the current state. Thirdly, issues with the policy gradient that could result from the above modifications are resolved by making the baseline zero-mean.
    - It seems like the authors do not realize this. The approach is cast as a practical method for improving the stability of training, but exactly what this method does and what it amounts to is not addressed by the authors.  The concerns from Action Editor ED9N are handwaved away.
- **Against episodic control.**  The approach of grouping similar trajectories together to adjust the reward is inherently a non-Markovian approach. This is in contrast to evaluating similar states to adjust the reward, as a baseline would, as well as episodic control, which employs states and not trajectories.
    - Within an MDP, where the state is a sufficient statistic for the expected reward-to-go, one should not expect this to yield any benefit.
    - Why do we see improvements in Section 4.4.1? In practice, the state might not be a sufficient statistic, leading to some unobserved confounding that yields non-Markovian behavior. This can be more robust to defending against that in practice.
    - However, employing a non-Markovian approach likely suffers from the curse of horizon, as is widely understood.
    - The benefit of richer information v.s. the drawbacks in horizon are not explored within the ablation study in section 4.4.1.
- Overall, this reflects on the quality of the exposition within the paper. Other issues related to this remain. For instance, which objective function was chosen for the policy in Algorithm 1 for the experiments? It is likely the PPO objective, but this has to be inferred from the writing within the experiments, as it is ambiguous whether "we conduct retraining experiments on the victim agents using the PPO algorithm" means retraining with the PPO objective or with the common implementation of the algorithm proper. Again, it is likely the former, but it is never clearly stated.
2. **On theory.**
- A proof of policy optimality is provided in the appendix. However, the proof is rather verbose, and adds complexity . The proof amounts to showing that the original reward is perturbed by a stochastic term with mean zero, so the Q-function remains the same, and the optimal policy must also remain the same. This can be expressed in a single line. I do not think that this is sufficient theoretical background per-se.
- Regarding the relation with a baseline, there is some opportunity for a derivation of how it differs from it here.
    - The episodic memory is not a TD(0) (or GAE(0)) critic, but there is an relation between this and REINFORCE with baseline, or GAE(1). To see this, consider the following:
        - Let $r'(s,a) := r(s) + \delta(s)$. It has been shown by the authors that $\mathbb{E} \delta(s) = 0$. Then, the policy gradient advantage estimate is $\sum_{l=0}^{\infty} \gamma^l r(s_{t+l}, a_{t+l}) - V(s_t)$.
        - The reward-shaped estimate is  $\sum_{l=0}^{\infty} \gamma^l (r(s_{t+l}, a_{t+l}) + \delta(s_{t+l})) = \sum_{l=0}^{\infty} \gamma^l r(s_{t+l}, a_{t+l}) - (\sum_{l=0}^{\infty} - \gamma^l \delta(s_{t+l})) =: \sum_{l=0}^{\infty} \gamma^l r(s_{t+l}, a_{t+l}) - B(s_{t:})$, where $ B(s_{t:}) := \sum_{l=0}^{\infty} - \gamma^l \delta(s_{t+l})$ is a function of the future trajectory.
        - This at first seems to invalidate the proof of an unbiased policy gradient, but given that $B(s_{t:})$ is mean-zero, this is resolved, and it follows that the policy gradient remains the same in expectation.
            - $\mathbb{E}[\sum_{l=0}^{\infty} \gamma^l (r(s_{t+l}, a_{t+l}) + \delta(s_{t+l})) ]= \mathbb{E}[\sum_{l=0}^{\infty} \gamma^l r(s_{t+l}, a_{t+l}) - (\sum_{l=0}^{\infty} - \gamma^l \delta(s_{t+l}))] = \mathbb{E}[\sum_{l=0}^{\infty} \gamma^l r(s_{t+l}, a_{t+l})] -  \sum_{l=0}^{\infty}\gamma^l \mathbb{E}[\delta(s_{t+l})] = \mathbb{E}[\sum_{l=0}^{\infty} \gamma^l r(s_{t+l}, a_{t+l})] - 0.$

**Requested Changes:**

Explore the relation to a baseline, the benefits of the non-Markovian trajectory-based episodic control v.s. Markovian state-based episodic control, and either beef up the theory (perhaps with the derivation provided) or be more upfront about the lack of contribution there. These are critical to securing my recommendation for acceptance.

---

> ### Author Response · Authors · 2025-08-29
> **Response to Reviewer FGt1**
>
> We appreciate the invaluable review, the enlightening thoughts, and the assistance on our theoretical improvement. Regarding Strength 1, the reviewer characterizes our contribution as “patterns vs. states.” To clarify, we do not consider “pattern” to be our contribution. We do not mention patterns in the Introduction; the first use appears in Section 3.2, where we state: “Therefore, we use state sequences, referred to as patterns, as the basis for performance evaluation in past episodes. Our choice of utilizing state sequences for performance evaluation aligns with existing research (Sutton & Barto, 2018; Li et al., 2023), which has shown that failures in adversarial games are often the result of a series of poor decisions rather than isolated states.” Our core contribution is the exploration of how a single state influences the final game outcome, together with a reward revision that injects outcome-aligned learning signals into the rewards. We respectfully ask the reviewer to reread the paper in light of this clarification and the responses that follow.
>
> ---
>
> 1. Relation to a critic, baseline, episodic control, etc.
>
> - **On baselines and critics.**
>
> >We agree that our episodic memory is similar to REINFORCE because both use episode-level returns and apply an injection step. However, unlike REINFORCE, which uses the trajectory’s own return and injects it into the gradient, our revision is based on a historical score which is a cross-episode statistic used to expose outcome-aligned reward signals for the states, and we inject it into the reward. REINFORCE is not a reward engineering method.
> >
> >After carefully reading the review, we acknowledge that a REINFORCE-like implementation of our method is possible, but we build our method as episodic-control-based reward engineering, which is an equally appropriate implementation for our goal since episodic control fits perfectly for our cross-episode historical evaluation, thus we only mentioned episodic control in our related work section.
>
> - **Against episodic control.**
>
> >We do not believe a “curse of horizon” applies to our method. First, our approach is reward engineering: the base learner is unchanged except that it receives shaped rewards, so the estimator’s effective temporal span is the same with or without our episodic memory. Second, the episodic memory is also not horizon-sensitive: the value assigned to a pattern is an episode-level return representing the game outcome, so each pattern in that episode receives the same value regardless of its temporal distance. Thus, we treat the pattern length as a hyperparameter and report the parameter analysis in the appendix. We can move that analysis into the main paper if the reviewer feels it would aid clarity.
>
> - **Overall**
>
> > Since our method is a reward engineering method, as shown in the workflow (Figure 1), there will be no change to the base learner except the stored rewards. The objective function relates only to the base learner, not to our episodic memory. Regarding “we conduct retraining experiments on the victim agents using the PPO algorithm,” although we implement our method on PPO, we separate the methods as ‘PPO’ and ‘our method’ in the paper. For better clarification, we will add “Our method only revises the environmental rewards without any modification to the base learner.” at the start (workflow) paragraph of Section 3. As for the pattern length, although we did not include the baselines in Figure 5 in Section 4.4.1, compared to Figure 2 in Section 4.2 (main results) we can see that even without the pattern, our method still greatly outperforms the baselines, which means the main effectiveness does not come from patterns vs. states but from our appropriate use of game-outcome-related reward-signal injection.
>
> ---
>
> 2. On theory.
>
> >We appreciate the proof from the REINFORCE perspective and agree it is very appropriate for a REINFORCE-like version of our method. In contrast, our actual implementation uses reward engineering, so the appendix adopts the inverse-RL convention of proving potential-based shaping. In inverse RL, potential-based shaping function is used to guarantee the unbiased policy optimality. It generally proposes a potential function $\phi(s_t)$ that differs across methods and takes $F(s,a,st’) = \gamma\*\phi(s’) - \phi(s)$ as the shaping value to make the added term $F$ have zero expected discounted sum. Because Action Editor ED9N cited the classical potential-shaping paper [1], we intentionally mirror that proof to demonstrate that our episodic feedback is a valid potential-based shaping function.
> >
> >[1] Andrew Y. Ng, Daishi Harada, and Stuart J. Russell. Policy Invariance Under Reward Transformations: Theory and Application to Reward Shaping, ICML 1999
>
> ---
>
> We hope our responses and clarifications have addressed your concerns. If there are any remaining questions, please let us know, and we will do our best to resolve them.

---

> > ### Comment · Reviewer_FGt1 · 2025-08-29
> >
> > Dear authors,
> >
> > **On patterns v.s. states**
> >
> > At no point during the review do I mention patterns. I believe you have confused my quote with that in your "changes since last submission" blurb relating to the old AE's comments. It is fair that my summary as to point 1 was rather terse, and I apologize for that.
> > - I believe it was clear during my review that I understood that your contribution was presented as reward shaping (or as the authors say within the response, "episodic-control-based reward engineering"). That it aims to capture winning v.s. losing states is implicit in me saying that it is similar to a baseline, which quite literally fits a value function, which is in turn exactly a score of states.
> > - I was saying that in contrast to episodic control, you score trajectories rather than states, which I believe is clear in Section 3.3.2. Within the paper, at no point do you actually score a state, except for when the length of $p_t$ is 1.
> > - I don't believe that going to Section 3.2 is unfair, as it relates to the core idea behind your contribution, while Section 3.3 relates to how you implement it.
> > - Furthermore, the exploration of how a single state affects the outcome is arguably the process of learning the value function. I don't believe this is what the authors aim to do.
> >
> > **On baselines and critics**
> >
> > The authors misinterpret my remark in W1. I did not say that your episodic memory is similar to REINFORCE. I said that your episodic memory is similar to the *use of a baseline within REINFORCE*. These are two very different things.
> > - Why do I talk about REINFORCE? The authors use PPO to train the reward-shaped agent, which is a policy gradient method. REINFORCE is arguably the simplest policy gradient method, and so exploring what it does in this case is not unfair. At no point did I say that REINFORCE is a reward engineering method (which is arguably a rather preposterous claim).
> > - Just because a baseline is used does not mean that one has to use REINFORCE. One can use a baseline within a PPO update step, which amounts to using GAE(1) within the PPO update.
> > - At no point do I suggest a REINFORCE-like implementation. What I suggested was that the method is related to the use of a baseline, which in turn was only brought up due to the previous AE's comments, which presumably the authors aimed to address.
> > - Do you mean that you "[] mentioned [only] episodic control", or "only mentioned episodic control"? The former means something quite different from the latter. The former suggests that the only thing mentioned was episodic control, but the latter means that you did not refer to episodic control anywhere else.
> >
> > **Against episodic control.**
> > Saying that the base learner is unchanged obscures the point. The expected rewards are the same, but it is possible that variance exponential in the length of the pattern could be injected into the rewards, therefore hindering learning. The original MDP and the reward-shaped MDP have the same transitions, states, actions, optimal policy, policy gradient on average, etc., but crucially not the same rewards, and not the same policy gradient variance.
> > - Why does the variance of the policy gradient matter? You use PPO, which is a policy gradient method. The variance in the rewards impacts the variance in the policy gradient. The variance in the policy gradient impacts the rate of convergence of PPO. If the variance in the policy gradient is exponential in the pattern length (which is quite possible in some hard instances), then the rate of convergence will be slowed by a factor exponential in the length of the pattern, or at least something quite nontrivial.
> > - To be fair, I did use the word "horizon" quite colloquially (because "curse of horizon" is the usual term for anything exponential in the length of a lookback or window or horizon, etc.). It is probably more accurate to say that this is the curse of the pattern length. The episodic memory is horizon-insensitive if we are referring to the MDP horizon proper, but it is not pattern-length insensitive, which is what I mean. Sorry for the confusion here.
> >
> > Hopefully this clarifies the review. Thank you for your continued engagement throughout the review process!

---

> ### Author Response · Authors · 2025-08-29
>
> Dear Reviewer FGt1,
>
> Thanks for your response. We apologize for habitually omitting “with baseline” when referring to REINFORCE with baseline. The reason we mentioned a “REINFORCE-like implementation” is because after reading your proof, we began considering whether one could inject an episodic-feedback–like term directly into the gradient to obtain the same effect without modifying the rewards. And regarding ‘we only mentioned episodic control’, this refers only to the related work section. We currently include episodic control there, but REINFORCE with baseline is not included.
>
> ---
>
> **On baselines and critics**
>
> > - Since our aim is to explore how a single state affects the outcome, after reconsidering the relationship between the historical score and the value function, we conclude that our historical score can be viewed as a special off-policy value function with discount factor 1 under a sparse-reward-like setting in which the reward is the total episode return. Because this special value function does not depend on the action, it can serve as a state-dependent baseline usable in REINFORCE with baseline and PPO with GAE(1).
> >
> > - On the other hand, the running mean of the average cumulative reward is also action-independent, so the difference between the special value function and the running mean is still an action-independent baseline, which means the episodic feedback can also be considered a baseline. Consequently, we acknowledge that the episodic memory can be treated as a baseline in Monte-Carlo settings like REINFORCE with baseline and PPO with GAE(1). We also notice that, although our episodic feedback can be considered a potential-based shaping function, a formal potential-based shaping term $F(s,a,s’)$ is not action-independent, whereas our episodic feedback is; therefore the similarity to a baseline exists only in our method. We think a similar discussion is necessary in our paper to clarify this point, and the proof from the REINFORCE-with-baseline perspective should also be included in the appendix.
>
> ---
>
> **Against episodic control**
>
> > - We agree with the idea that “The original MDP and the reward-shaped MDP have the same transitions, states, actions, optimal policy, policy gradient on average, etc., but crucially not the same rewards, and not the same policy gradient variance.” In our previous submission we wrote: “Generally, reward engineering methods rely heavily on empirical insights and experimental validation, as theoretical analysis of how reward revisions influence the training process is inherently challenging.” in the discussion. The specific challenge we meant is exactly the reward variance issue: how changes in reward (policy gradient) variance affect the learning process even when everything else is unchanged. We do not yet have a complete theoretical explanation and this is basically the “future work” we mentioned in the discussion.
> >
> > - We appreciate the clarification of “horizon.” Previously we treated pattern length k mainly as a parameter analysis problem. In early experiments on YouShallNotPassHumans and KickAndDefend, we observed a noticeable decline at k = 10 or 20 (while still outperforming baselines) and simply attributed this to information loss from encoding longer sequences into an abstract vector. After reconsidering “horizon,” we now think the decline may reflect the curse of pattern length. However, our understanding of the curse of pattern length is different, which is increasing the pattern length exponentially enlarges the pattern space (for example, if a single state can be matched into $N$ types, then for a pattern, there are $N^k$ different matches), reducing matches per pattern and thus cause the learning signal (historical score) to be noisier. Incorporating such learning signals as the reward variance may harm the convergence of the base learner.
> >
> > - Besides, for “The benefit of richer information v.s. the drawbacks in horizon are not explored within the ablation study in section 4.4.1.”, we are wondering whether incorporating experiments with more pattern length like 7, 10, 20 to Section 4.4.1 will be helpful to understand the trade-off.
>
> ---
>
> Thank you again for your insightful review and follow-up. We look forward to your further feedback.

---

### Review · Reviewer_HGpX · 2025-08-28

**Summary Of Contributions:**

The authors propose a new approach for training black-box adversarial agents, which involves using historical state information to alter rewards to improve the adversarial policy. This generally resulted in beating the baseline adversarial policies.

**Audience:**

No

**Audience Explanation:**

I believe individuals in TMLR's audience may not be interested in this paper for two main reasons:
* The proposed approach, while getting strong results, was relatively straightforward and intuitive, and doesn't really bring many new learnings to the table. The proposed approach may simply outperform due to more compute/complexity.
* The paper was extremely difficult to follow, due to the reasons described in the requested changes below. If the paper had been more digestible, I would feel the audience may be more interested in the paper.

**Claims And Evidence:**

Yes

**Claims Explanation:**

The claims regarding improvements over existing baselines for adversarial policies are indeed confirmed by convincing and clear evidence. Figures 2-6 demonstrate strong results, particularly Figures 2 and 3. The proposed algorithm generally outperforms existing baselines more often, although on certain select environments, this is not the case.

**Requested Changes:**

- The writing style and explanation of concepts/ideas is generally very difficult to understand. This made it challenging to understand the method, introduction, related works, etc (with the methodology section being extremely difficult to understand). This is critical. There are numerous mistakes in flow even in the abstract, which makes the paper difficult to understand even at a high level. Although I generally don't believe grammar/flow should affect the review of a paper, I found it to really impact the readability of this paper.
- There are many occasions where details are inserted in seemingly random places in the paper that do not make logical sense. I highly recommend the authors rework the entire flow of the paper to be more story-like and follow more logically. This is critical. E.g. I did not find the discussion to contain relevant information to the discussion (i.e. it discussed experimental details).
- A lot of the claims made in the paper are incorrect or misleading, these should be adjusted and are critical:
	- "It has been **proved** that deep reinforcement learning (DRL) policies are vulnerable to adversarial attacks"
		- There is no hard proof just strong evidence, it has been shown not proved
	- "It is also important to note that our method is scalable and can be applied to various DRL algorithms."
		- There needs to be more evidence to make such claims
- Should "Object Function" in Figure 1 be objective function? This Figure is unclear to me as a consequence and as a whole. A step by step explanation of this Figure in the caption would be helpful.
- The baselines compared to, from 2019-2021, are not modern baselines, the authors should include more modern baselines.
- There are minor formatting issues with many of the citations.

---

> ### Author Response · Authors · 2025-08-29
> **Response to Reviewer HGpX**
>
> We appreciate the time and effort the reviewer devoted to our work and thank them for pointing out our mistype on “objective function” in our figure. However, we do not find the rest of the review appropriate: it states that our writing is difficult to understand and lacks logical coherence without providing concrete examples or questions. Without specific pointers, we cannot identify what to clarify or revise.
>
> ---
>
> A lot of the claims made in the paper are incorrect or misleading, these should be adjusted and are critical:
>
> - There is no hard proof or strong evidence for "It has been proved that deep reinforcement learning (DRL) policies are vulnerable to adversarial attacks."
>
> > There are plenty of previous works that show DRL is vulnerable to adversarial attacks, and this is, to some extent, a common knowledge in the field. We reference 5 papers [1–5] as examples.
> >
> >[1] Huang, S., Papernot, N., Goodfellow, I., Duan, Y., and Abbeel, P. Adversarial attacks on neural network policies. arXiv preprint arXiv:1702.02284, 2017.
> >
> >[2] Behzadan, V. and Munir, A. Vulnerability of deep reinforcement learning to policy induction attacks. In International Conference on Machine Learning and Data Mining in Pattern Recognition, pp. 262–275. Springer, 2017.
> >
> >[3] Lin, Y.-C., Hong, Z.-W., Liao, Y.-H., Shih, M.-L., Liu, M.-Y., and Sun, M. Tactics of adversarial attack on deep reinforcement learning agents. arXiv preprint arXiv:1703.06748, 2017.
> >
> >[4] Pattanaik, A., Tang, Z., Liu, S., Bommannan, G., and Chowdhary, G. Robust deep reinforcement learning with adversarial attacks. In Proceedings of the 17th International Conference on Autonomous Agents and MultiAgent Systems, pp. 2040–2042. International Foundation for Autonomous Agents and Multiagent Systems, 2018.
> >
> >[5] Xiao, C., Pan, X., He, W., Peng, J., Sun, M., Yi, J., Li, B., and Song, D. Characterizing attacks on deep reinforcement learning. arXiv preprint arXiv:1907.09470, 2019.
>
> - There needs to be more evidence to make such claims for "It is also important to note that our method is scalable and can be applied to various DRL algorithms."
>
> > The reward engineering methods do not change the base learner so they can be applied to different DRL learner unless the algorithm also modifies the rewards. We believe our experiments of our method implementing on different baselines can support for this claim.
>
> ---
>
> The baselines compared to, from 2019-2021, are not modern baselines, the authors should include more modern baselines.
>
> > Although the baselines are from 2019-2021, Gleave et.al. (2019) is the basic adversarial training method. To our best knowledge, Guo et.al and Wu et.al. (2021) are still state-of-the-art methods in the subfield as there are no more recent methods which outperforms them in all four environments.

---

> > ### Comment · Reviewer_HGpX · 2025-08-29
> >
> > I am happy to provide more locations where the grammar or logical coherence is not clear:
> > - "Our method extracts the historical evaluations for states from historical experiences with an episodic
> > memory, and then incorporating these evaluations into the rewards with our proposed reward revision method to improve the adversarial policy optimization."
> > - "It has been proved..."
> > - "In recent researches, episodic control (Lengyel & Dayan, 2007) has been proved effective on modifying environmental rewards to address sample inefficiency in various tasks, such as multi-agent tasks..."
> > - The final paragraph in the related works seems to not flow as a traditional related works paragraph, where the authors spend the majority of the paragraph discussing motivation/approach and  experimental results.
> > - I still found the caption of Figure 1, arguably the most important figure, to be lacking.
> > - "We use the same setting as Gleave et al. and assumes that our threat model has control over the adversarial agent and black-box access to the information of the victim agent."
> >
> > These are all in the first 3.5 pages. I found many more but I strongly believe that it's fair to say the flow, logical coherence, and grammar make the paper challenging to follow/understand. I also discussed issues with sections such as the Discussion section, which seem to contain relevant information.
> >
> > Again, I want to emphasize that using words like "prove" can be dangerous in science. Generally, things are not proven in science, and casually using the term is simply too strong when a few scientific papers have been cited. While there may be strong evidence for something, proving something is a **much** stronger statement.
> >
> > Regarding the point on scalability, I did not understand the author's response. Showing scalability often involves testing models at various scales (i.e. data, FLOPs, parameters, etc); and comprehensively comparing these to baseline scalable approaches. Without these results, making a claim like "our approach is scalable" is completely unfounded.
> >
> > A lot of my review still seems unaddressed, such as the information above, the citations, I found the explanation of modern baselines lackluster, and the flow.

---

> > > ### Author Response · Authors · 2025-08-31
> > >
> > > Dear Reviewer HGpX,
> > >
> > > Thanks for your further clarifications.
> > >
> > > ---
> > >
> > > - "Our method extracts the historical evaluations for states from historical experiences with an episodic memory, and then incorporating these evaluations into the rewards with our proposed reward revision method to improve the adversarial policy optimization."
> > >
> > > > We will change “incorporating” to “incorporates”.
> > >
> > > - "It has been proved..."
> > >
> > > > We acknowledge that “prove” can be a strong word. “Prior work shows that DRL policies can be vulnerable to adversarial attacks (Huang et al., 2017; Kos & Song, 2017).” should be a more precise expression. We will also change the “prove” problem in the abstract.
> > >
> > > - "In recent researches, episodic control (Lengyel & Dayan, 2007) has been proved effective on modifying environmental rewards to address sample inefficiency in various tasks, such as multi-agent tasks..."
> > >
> > > > We will change it to "In recent **research**, episodic control (Lengyel & Dayan, 2007) **shows its effectiveness** on modifying environmental rewards to address sample inefficiency in various tasks, such as multi-agent tasks…”.
> > >
> > > - The final paragraph in the related works seems to not flow as a traditional related works paragraph, where the authors spend the majority of the paragraph discussing motivation/approach and experimental results.
> > >
> > > > While we think that providing the differences between our method and existing work will help readers understand our improvement, the last paragraph in the “Reward Engineering” section is unnecessarily long for a related work section. We will simplify the whole paragraph as follows:
> > > >
> > > >“Our method adopts episodic control as the base since it shares a similar idea of using historical experience to guide learning. Unlike methods that directly add statistics from episodic memory to the reward, we apply a conditional revision aligned with outcome-aligned reward signals and rely on a neural episodic memory for efficiency. Compared with a state-of-the-art episodic control method NECSA (Li et al., 2023), our approach performs better with less time cost (Sec. 4.3).”
> > >
> > > - I still found the caption of Figure 1, arguably the most important figure, to be lacking.
> > >
> > > > We included a step-by-step explanation of the workflow figure at the beginning of Section 3. A similar explanation will be added to the figure caption.
> > >
> > > - "We use the same setting as Gleave et al. and assumes that our threat model has control over the adversarial agent and black-box access to the information of the victim agent."
> > >
> > > > We will change “assumes” to “assume”.
> > >
> > > ---
> > >
> > > Regarding the point on scalability, what we want to emphasize is applicability across base learners, not the kind of scalability you mentioned. We will therefore change “It is also important to note that our method is scalable and can be applied to various DRL algorithms.” to “It is also important to note that our method can be applied to various DRL algorithms.”
> > >
> > > For the Discussion section, we can move the StarCraft II part into the Experiments section, but we think the discussion of implementing our method on other DRL algorithms should remain here. This capability is not something we specially designed; it is a general property of reward engineering methods, and we believe it is worth noting.
> > >
> > > Regarding modern baselines, we apologize that we did not clearly explain them in our last response. After Guo et al. and Wu et al., we did not find much recent work on adversarial training for two-player competitive games. The only method we consider state-of-the-art is Bui et al., which we already mentioned in Related Work. However, it depends on additional information (i.e., it is not a fully black-box method) compared with Gleave et al., Guo et al., Wu et al., and ours. Therefore, we do not consider it an appropriate baseline for our paper.
> > >
> > > For the citations, could you please be more specific about the formatting issues? We use “\bibliographystyle{tmlr}” and paste the BibTeX from Google Scholar into the .bib file without modification. We will fix any items you point out.
> > >
> > > ---
> > >
> > > We hope our responses and clarifications have addressed your concerns. If there are any remaining questions, please let us know, and we will do our best to resolve them.

---

### Author Response · Authors · 2025-09-10

Dear Reviewers,

We thank you for your constructive feedback and have uploaded a revised manuscript. In this revision, we made the following changes based on your suggestions:

- Fix the grammar and clarity issues pointed out by Reviewer HGpX and Reviewer AeY8.
- Replace "prove" and "scalable" with more suitable phrasing.
- Condense the final paragraph in Related Work (Section 2.2).
- Expand the caption for Figure 1 to include a step-by-step explanation of our method's workflow.
- Add our hardware details to the experimental setup in Section 4.1.
- Move the StarCraft II experiments from the Appendix to the main body (new Section 4.4, the ablation study becomes Section 4.5).
- Add an analysis for 'the curse of pattern length' in ablation study of pattern (Section 4.5.1), with a performance comparison across six pattern lengths (k=1,3,5,7,10,20).
- Include a new discussion in Section 5 regarding our episodic feedback's functional similarity to a value baseline.
- Clarify the discussion on future work in Section 5, identifying the reward variance issue as a key challenge.

Regarding the additional parameter analysis for group size, we are currently running experiments with group-200 and group-300. As the experiments could not be completed before the discussion phase deadline, we have uploaded the manuscript without them for now. We will add these results to the final version of the paper once all experiments are finished.

We hope these revisions have addressed your concerns and thank you again for your valuable reviews.

---

> ### Author Response · Authors · 2025-09-26
>
> Dear Reviewers,
>
> We have included the results for group size in the latest version.

---

### Decision · Action_Editor_kGvZ · 2025-10-06

**Recommendation:** Accept with minor revision

**Additional Comments:**

Currently there are still gaps between theories and practices. Theories only discuss why this method can be helpful but not necessarily mean the proposed method should be helpful. It would be better if authors can provide more clear justifications on why their methods should work under their setup, even under strong assumptions.

**Audience:**

Yes

**Audience Explanation:**

Although this submission focused on restricted area on improving adversarial training of RL agents, some insights (and the revisitation of the historical methods) as well as the experimental setup may be helpful for more general research on adversarial robustness and RL.

**Claims And Evidence:**

Yes

**Claims Explanation:**

The authors have effectively addressed most prior concerns, and the method's justification is sound. However, the reviewers and I still find the connection between the theoretical background and the practical method unclear. The authors should make this link more direct, moving beyond intuition to show how theory explicitly guides their proposed approach.

---

> ### Author Response · Authors · 2025-10-23
>
> Dear reviewers and action editor,
>
> We sincerely thank the reviewers and the action editor for your thoughtful and constructive feedback to help us improve the clarity and quality of our manuscript. We have uploaded the camera ready version and the revision includes:
>
> - We add the link to our code repo in the abstract.
>
> - We rewrite Section 3.2.2 to improve the empirical clarification of episodic feedback. We understand the requested change is mainly about the theory, but ‘why our method should be effective’ is highly related to ‘how reward variances improve the learning process’, and our current theory cannot solve that. Therefore, we have tried our best to provide empirical explaination of the effectiveness of our method.
>
> We would be happy to make further adjustments if needed.